**EMBO**
*reports*

# MEX3A regulates *Lgr5*⁺ stem cell maintenance in the developing intestinal epithelium

Bruno Pereira[1,2,*] (iD), Ana L Amaral[1,2,†], Alexandre Dias[1,2,†], Nuno Mendes[1,2], Vanesa Muncan[3], Ana R Silva[1,2], Chantal Thibert[4] (iD), Anca G Radu[4], Leonor David[1,2,5], Valdemar Máximo[1,2,5], Gijs R van den Brink[3,6], Marc Billaud[7] & Raquel Almeida[1,2,5,8,**] (iD)

## Abstract

Intestinal stem cells (ISCs) fuel the lifelong self-renewal of the intestinal tract and are paramount for epithelial repair. In this context, the Wnt pathway component LGR5 is the most consensual ISC marker to date. Still, the effort to better understand ISC identity and regulation remains a challenge. We have generated a *Mex3a* knockout mouse model and show that this RNA-binding protein is crucial for the maintenance of the *Lgr5*⁺ ISC pool, as its absence disrupts epithelial turnover during postnatal development and stereotypical organoid maturation *ex vivo*. Transcriptomic profiling of intestinal crypts reveals that *Mex3a* deletion induces the peroxisome proliferator-activated receptor (PPAR) pathway, along with a decrease in Wnt signalling and loss of the *Lgr5*⁺ stem cell signature. Furthermore, we identify PPARγ activity as a molecular intermediate of MEX3A-mediated regulation. We also show that high PPARγ signalling impairs *Lgr5*⁺ ISC function, thus uncovering a new layer of post-transcriptional regulation that critically contributes to intestinal homeostasis.

**Keywords** intestinal homeostasis; LGR5⁺ intestinal stem cells; mouse model; PPARγ pathway; RNA-binding protein MEX3A
**Subject Categories** Development; Signal Transduction; Stem Cells & Regenerative Medicine

## Introduction

Stem cells ensure homeostasis throughout life, and the intestinal tract is the organ that best displays this extraordinary ability. It is the fastest self-renewing tissue in mammals, as epithelial turnover takes about 3–5 days to be completed [1]. Intestinal stem cells (ISCs), located in mucosal invaginations known as crypts of Lieberkühn, control this process. ISCs generate proliferating transit-amplifying (TA) cells that after several divisions commit into differentiated cell types. These mainly include nutrient-absorbing enterocytes, mucous-producing goblet cells, hormone-secreting enteroendocrine cells and chemosensing tuft cells; all located along luminal protrusions called villi. Paneth cells, which play a role in innate immunity by synthesizing antimicrobial peptides, are unique because they are the only differentiated cell type that stays in the crypt with a long residence time of 6–8 weeks.

The location and nature of ISCs remains a matter of intense debate. Expression profile and lineage tracing experiments performed over the last decade have led to the identification of at least two distinct ISC populations: crypt base columnar (CBC) cells and +4 cells. CBC cells, wedged between Paneth cells at the base of the crypt, are actively cycling and sensitive to radiation-induced damage. The Wnt/β-catenin pathway target gene *Lgr5* is one of the most reliable CBC markers [2], and single *Lgr5*⁺ cells originate organoids or "mini-gut"-like structures *ex vivo* comprising all intestinal cell types [3]. Additional markers for CBC stem cells include *Olfm4* and *Ascl2* [4,5]. The +4 cells, located on average 4 positions above the crypt bottom and over the uppermost Paneth cell, are relatively quiescent and resistant to radiation, expressing markers such as *Bmi1*, *Hopx*, *Lrig1* and *Tert* [6–10]. These populations are not hard-wired, and intricate patterns of interconversion seem to exist between them, both under homeostasis and in response to injury [10–13]. Moreover, it has been shown that *Dll1*⁺

1 i3S - Institute for Research and Innovation in Health (Instituto de Investigação e Inovação em Saúde), University of Porto, Porto, Portugal
2 IPATIMUP - Institute of Molecular Pathology and Immunology, University of Porto, Porto, Portugal
3 Department of Gastroenterology and Hepatology, Amsterdam UMC, Tytgat Institute, University of Amsterdam, Amsterdam, The Netherlands
4 Institute for Advanced Biosciences, INSERM U1209, CNRS UMR5309, University Grenoble Alpes, Grenoble, France
5 FMUP-Faculty of Medicine, University of Porto, Porto, Portugal
6 Medicines Research Center, GSK, Stevenage, UK
7 Clinical and Experimental Model of Lymphomagenesis, INSERM U1052, CNRS UMR5286, Centre Léon Bérard, Université Claude Bernard Lyon 1, Centre de Recherche en Cancérologie de Lyon, Lyon, France
8 Biology Department, Faculty of Sciences, University of Porto, Porto, Portugal
*Corresponding author. Tel: +351 2260 74900; E-mail: bpereira@ipatimup.pt
**Corresponding author. Tel: +351 2260 74900; E-mail: ralmeida@ipatimup.pt
†These authors contributed equally to this work

[14], *Atoh1*[15–17] or *Prox1*[18] secretory progenitor cells, *Alpi*[+] enterocyte precursors [19] and even subsets of committed *Lyz1*[+] Paneth cells [20,21] can all present stem cell activity, illustrating the level of cellular plasticity in the gut.

Most studies focusing on ISC biology have been performed using adult animal models, when intestinal homeostasis is stably established, whereas little is known about one of the most active phases of this cellular population, which is the postnatal period between birth and weaning [22]. Newborn mice possess an immature epithelium that contains villi formed in late embryogenesis but lacks crypts. Instead, a small number of *Lgr5*[+] progenitors are restricted to basal regions between villi called intervillus domains [23,24]. The foetal LGR5 progeny is, by itself, insufficient to sustain intestinal growth during initial development [25], suggesting the presence of *Lgr5*[−] precursors in the earliest phases of postnatal development. To ensure proper epithelial cell renewal, the functional crypt-villus axis must develop rapidly after birth, which is accomplished by an initial expansion of the entire stem cell pool via symmetric divisions, followed by a sharp transition to TA cell production via asymmetric divisions [26]. In this phase, the intestinal epithelium is not in a steady-state condition, as in the adult counterpart, because cell production rate necessarily exceeds cell loss. Concomitant with these structural alterations, the epithelium undergoes major biochemical changes to support the dietary adaptation associated with the suckling-to-weaning transition [27,28]. The exact mechanisms involved in initiating postnatal expansion of ISCs and controlling its timing are unknown, but probably involve an intrinsic genetic programme and microenvironmental factors (e.g. circulating hormones, diet, microbiota), as well as molecular determinants mediating communication between both.

It is becoming clear that RNA-binding proteins (RBPs) and post-transcriptional mechanisms underpin stem cell fate decisions in response to different stimuli [29,30]. In this regard, we have been studying the evolutionarily conserved MEX-3 family of RBPs. Vertebrates have four homologous genes designated *MEX3A* to *MEX3D* encoding related proteins with two K Homology (KH) domains that provide RNA-binding capacity [31], and a Really Interesting New Gene (RING) C-terminal domain, which possibly mediates E3 ubiquitin ligase activity [32]. The different MEX-3 members are post-transcriptional regulators involved in embryonic patterning [33], pluripotency [34], fertility [35], immune responses [36], metabolism [37] and cancer [38]. Our previous work demonstrated that MEX3A overexpression is associated with stemness features in gastrointestinal cancer cell lines, including higher expression of the ISC markers *LGR5*, *BMI1* and *MSI1* [39]. In agreement, *Mex3a* mRNA is part of the *Lgr5*[+] signature [40], and its expression upregulated in a mouse model overexpressing *MSI1* in the intestinal epithelium [41]. Recently, *Mex3a* expression was observed in a subset of *Lgr5*[+] cells around the crypt +3/+4 cell position [42]. However, none of the previous studies has functionally addressed if MEX3A is essential in the context of ISCs.

In the present study, through the detailed characterization of a novel mouse model with a *Mex3a* deletion, we show for the first time that MEX3A is critical for the *in vivo* maintenance of the *Lgr5*[+] ISC pool. *Mex3a* null mice exhibit growth retardation and postnatal mortality due to impaired epithelial turnover, underlined by a dramatic decrease in *Lgr5*[+] ISCs and TA cells. Additionally, we provide evidence that *Mex3a* deletion leads to the aberrant activation of the peroxisome proliferator-activated receptor (PPAR) signalling pathway and establish PPARγ signalling as a molecular intermediate of MEX3A-mediated regulation. Our data uncover a new regulatory mechanism in ISCs of the developing gut with implications for intestinal homeostasis.

# Results

### Characterization of *Mex3a* expression pattern in murine tissues

We started by examining the *Mex3a* expression pattern among major organs in the mouse during postnatal development. By *in situ* hybridization (ISH), we determined that *Mex3a* mRNA was highly expressed in the thymus, moderately expressed in the brain and gut, lowly expressed in the stomach and skin, and absent from the heart, liver and lung (Fig EV1). In the intestinal tract, *Mex3a* transcripts were concentrated at the base of the small intestine and colonic crypts (Fig EV1, small intestine and colon inserts). In the skin, *Mex3a* mRNA was present in hair follicle-related structures only (Fig EV1, skin insert). The precise compartmentalization of *Mex3a* expression in stem cell niches of two of the most rapidly self-renewing mammalian organs, intestine and skin, suggested an *in vivo* function for MEX3A in stem cell biology.

### *Mex3a* null mice exhibit growth retardation and postnatal mortality

To address the physiological role of *Mex3a*, we characterized mice with a targeted intragenic deletion of the *Mex3a* locus coding sequence, developed under the framework of the INFRAFRONTIER-I3 European Research Infrastructure [43]. The initial deletion cassette consisted of a *LacZ* reporter cDNA followed by a *floxed* promoter-driven neomycin (*Neo*) resistance gene (Fig 1A). The *Mex3a*[tm1(KOMP)Vlcg] strain was generated and crossed with the epiblast-specific *Meox2*[+/Cre] deleter strain for removal of the *Neo* gene, giving rise to *Mex3a*[+/−] heterozygous mice. Heterozygous breeding schemes were set up and *Mex3a*[−/−] homozygous mutant animals were born, although not at the expected Mendelian frequency (13% versus 25% expected, Fig 1B), indicating the occurrence of embryonic lethality.

*Mex3a* knockout (KO) pups displayed severe growth retardation, presenting smaller size and weight when compared to *Mex3a*[+/−] heterozygous and wild-type (WT) siblings (Fig 1B and C). At postnatal day (P)15, *Mex3a* KO animals had an average weight of $4.00 \pm 0.16$ g (mean ± standard error, $n = 26$), compared to $6.57 \pm 0.14$ g ($n = 43$) of WT mice, a 39% weight difference (Fig 1C). This trait persists in mutant mice that reach adulthood. Both adult males and females are fertile and seem to have a normal lifespan. However, around 60% of the mutant animals developed a clinical condition characterized by gradual weight loss, signs of dehydration and progressive lethargy, between P15 and P21, eventually culminating in death (Fig 1D). These pups displayed normal suckling behaviour, as judged by the presence of milk in their stomachs (Appendix Fig S1). Macroscopic assessment of major organs indicated no gross morphological abnormalities, with the notable exception of the intestinal tract (Fig 2A and Appendix Fig S1). This subset of *Mex3a* null mice presented a translucent and air-filled gut tube, particularly evident in the ileum, caecum and colon (Fig 2A).

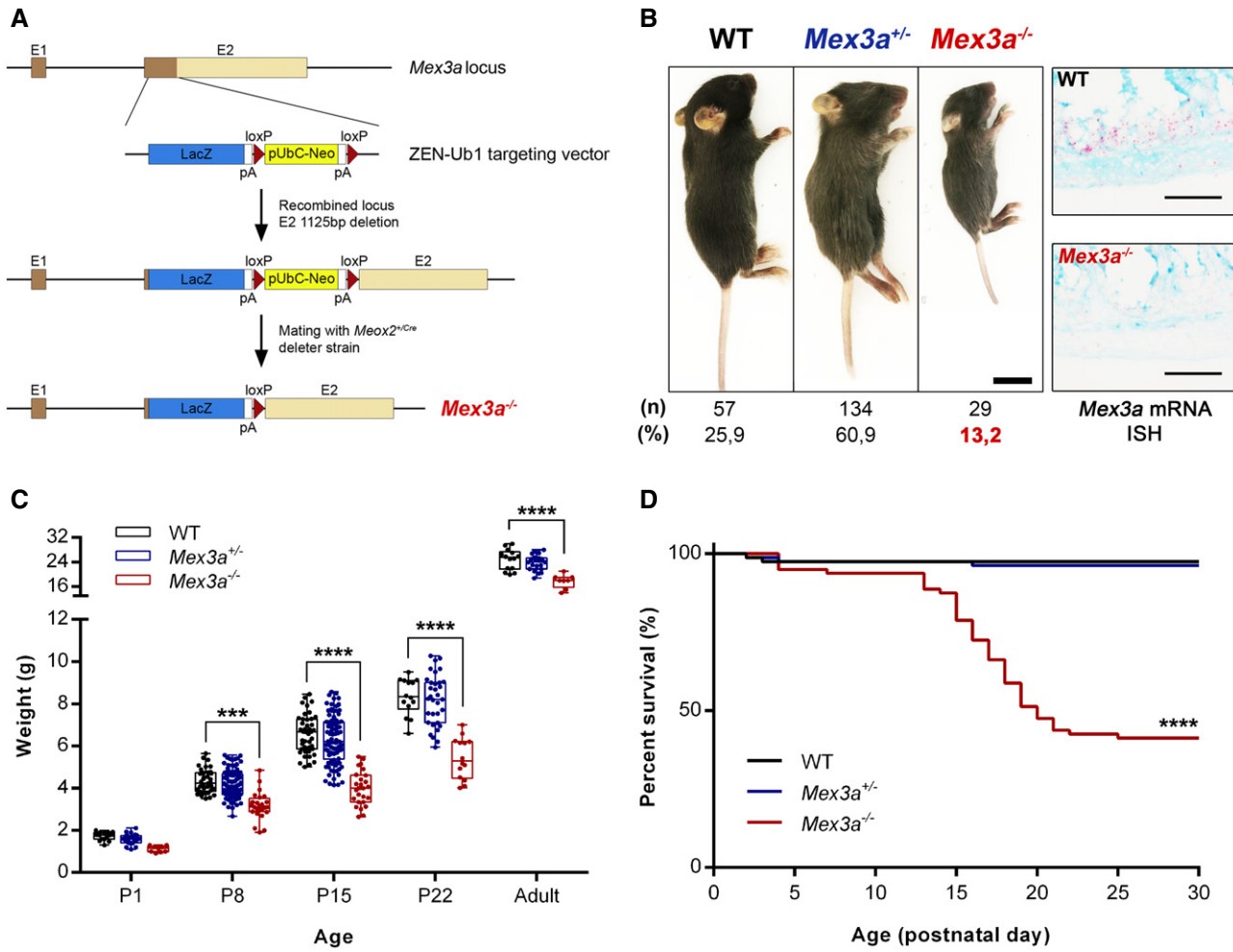

**Figure 1. *Mex3a* knockout mice exhibit smaller size and postnatal lethality.**

A   Scheme of the targeting vector for intragenic deletion of the mouse *Mex3a* gene. The insertion of the Velocigene cassette ZEN-Ub1 created a deletion of 1,125 bp in exon 2 of the *Mex3a* locus.

B   Representative images of the size of *Mex3a* mutant mice and control littermates at postnatal day (P)15. Scale bar, 1 cm. Genotypes were confirmed by *Mex3a* mRNA ISH in intestinal tissue (right panels). Scale bars, 50 μm. The offspring number (n) and observed genotype frequencies (%) resulting from heterozygous crosses are indicated below.

C   Absolute weight of *Mex3a* KO mice and control littermates at different ages. Data are represented in a box-and-whisker plot as mean (middle line) with the minimum and maximum distribution values. Each point depicts one animal (WT: P1, *n* = 15; P8, *n* = 36; P15, *n* = 43; P22, *n* = 14; Adult, *n* = 14; *Mex3a^+/−^*: P1, *n* = 24; P8, *n* = 90; P15, *n* = 90; P22, *n* = 37; Adult, *n* = 21; *Mex3a^−/−^*: P1, *n* = 9; P8, *n* = 24; P15, *n* = 26; P22, *n* = 14; Adult, *n* = 8) ***$P$ = 0.0003, ****$P$ < 0.0001, two-way ANOVA test.

D   Kaplan–Meier survival curves for the different *Mex3a* genotypes (*n* = 80 for each genotype) ****$P$ < 0.0001, log-rank (Mantel–Cox) test.

The overlap between the observed phenotype and the occurrence of important events in murine intestinal ontogenesis during this developmental time-window prompted us to focus on the effect of *Mex3a* deletion specifically in the intestinal epithelium.

### *Mex3a* deletion severely impairs normal intestinal crypt development

Histological analysis of haematoxylin and eosin (H&E)-stained paraffin sections of the intestinal tissue of *Mex3a* KO animals revealed severely altered crypt-villus architecture, particularly in the distal small intestine, with the presence of a reduced number of crypts and of smaller size when compared to littermate or age-

matched control animals (Fig 2B). Average crypt depth in *Mex3a* KO mice was 17.66 ± 1.33 μm compared with 30.05 ± 1.20 μm in WT mice (Fig 2C). Intestinal villi of mutant animals were also shorter, with a mean height of 98.86 ± 6.60 μm compared with 131.50 ± 10.82 μm in control mice (Fig 2D). A similar trend was observed for the proximal small intestine (Appendix Fig S2).

To explore the consequences of disrupting the *Mex3a* gene for intestinal differentiation, we assessed the presence of the main epithelial cell types using specific markers. The goblet and enteroendocrine cell lineages were not significantly modified as determined by alcian blue–periodic acid–Schiff (AB-PAS) reaction (Fig EV2A) and synaptophysin (SYP) staining (Fig EV2B), respectively, with a normal distribution of scattered cells amidst the mucosa.

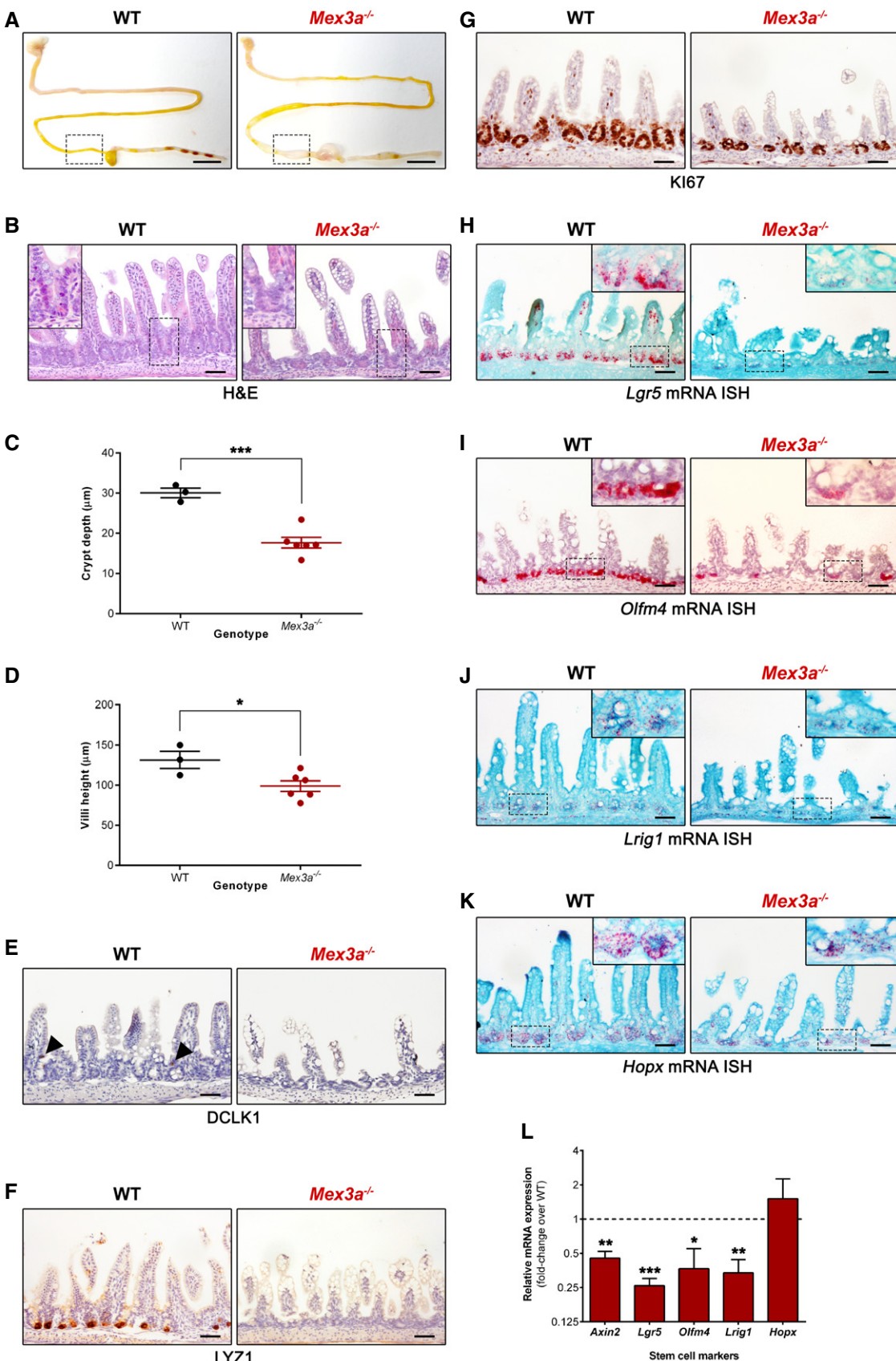

**Figure 2.**

**Figure 2.** *Mex3a* deletion leads to loss of *Lgr5*⁺ stem cells and impairs normal intestinal crypt development.

A    Macroscopic assessment of the gastrointestinal tract of *Mex3a* KO and WT mice. These images are representative of the phenotype observed in mutant animals euthanized at different postnatal days. Boxed areas depict the distal small intestinal section (ileum) used for subsequent immunohistochemical analyses.

B    H&E staining of a *Mex3a* KO and WT littermate at P19. Inserts depict high magnification of the boxed areas.

C, D   Average crypt depth (C) of WT animals ($n = 3$, > 40 crypts counted per animal) and *Mex3a* mutants ($n = 6$, > 20 crypts counted per animal). Also shown is the average villi height (D) of WT animals ($n = 3$, > 25 villi counted per animal) and *Mex3a* mutants ($n = 6$, > 15 villi counted per animal). Data are represented as mean ± standard error between P16 and P18. *$P = 0.03$, ***$P = 0.0006$, Student's *t*-test.

E–G   Immunohistochemistry staining for (E) the tuft cell marker DCLK1 (black arrowheads indicate positive cells), (F) the Paneth cell marker LYZ1 and (G) the proliferation marker KI67.

H–K   mRNA ISH staining for the stem cell markers (H) *Lgr5*, (I) *Olfm4*, (J) *Lrig1* and (K) *Hopx* in ileal sections of *Mex3a* mutants and littermate controls at P19. Inserts depict high magnification of the boxed areas.

L    qPCR analysis of the expression level of the indicated stem cell markers in freshly isolated crypt fractions from *Mex3a* KO and WT animals ($n = 3$ of each genotype). Data are represented as mean fold-change plus standard error in *Mex3a* KO mice relatively to WT animals (dashed line). *$P = 0.0208$, **$P < 0.01$, ***$P = 0.0007$, Student's *t*-test.

Data information: Scale bar (A), 1 cm; all other scale bars, 50 μm.

Enterocytes were equally detected along the lining of the villi by expression of villin (VIL1), a structural marker for the apical brush border (Fig EV2C). On the other hand, tuft cells were almost entirely absent from the *Mex3a* mutant mice intestine as observed with doublecortin-like kinase 1 (DCLK1), a specific tuft cell marker (Fig 2E). Since we previously demonstrated in cancer cell lines that MEX3A post-transcriptionally represses the expression of caudal-type homeobox 2 (CDX2) protein [39], a master regulator of intestinal maturation, we assessed CDX2 expression in the mutant animals. Although not consistently observed, CDX2 levels seemed increased in the incipient crypt compartment of *Mex3a* KO mice when compared with WT (Fig EV2D). Considering the hypothesis of a premature maturation of the intestinal epithelium, we tested the expression of sucrase-isomaltase (SIS, Fig EV2E) by immunohistochemistry. Normally, this enzyme starts to be expressed during the suckling-to-weaning transition and accompanies the change to solid food, being involved in the digestion of complex carbohydrates [27,28]. A premature expression of SIS was not detected in *Mex3a* KO mice at P15. As expected, the suckling period-specific enzyme argininosuccinate synthetase 1 (ASS1), which is involved in arginine synthesis, was broadly present in both mutant and control animals in the same period (Fig EV2F).

Regarding crypt cell populations, we detected a strong inhibition of Paneth cell expansion as assessed by lysozyme (LYZ1) staining in the *Mex3a* KO intestine (Fig 2F). We equally detected a significant lower number of proliferating cells identified by KI67 staining in the *Mex3a* KO intestine (Fig 2G). The average number of KI67⁺ cells per crypt was $6.56 \pm 1.39$ compared with $12.60 \pm 1.07$ in WT mice, a 48% difference (Appendix Fig S3). Collectively, these data demonstrate that MEX3A is not directly involved in the intrinsic mechanisms associated with intestinal maturation but is necessary for normal crypt development.

## Loss of *Lgr5*⁺ stem cells in the small intestine of *Mex3a* KO mice

Given the pronounced atrophy detected at the crypt level, we decided to investigate whether *Mex3a* deletion influenced ISCs responsible for maintaining small intestine epithelial homeostasis. For that, we performed ISH for the specific CBC cell marker *Lgr5* [2]. Strikingly, we observed a strong reduction in *Lgr5* mRNA levels in the intestinal crypts of *Mex3a* KO animals when compared to WT (Fig 2H). This reduction could be the result of either diminished *Lgr5* gene transcription or loss of the *Lgr5*-expressing ISCs. To

discriminate between the two, we performed ISH against *Olfm4* mRNA, another standard and highly robust marker of CBC cells [4]. *Mex3a*-deleted crypts also exhibited a very low amount of *Olfm4* transcripts (Fig 2I), thus reinforcing the evidence for *Lgr5*⁺ ISCs loss. We also assessed the mRNA expression level of the +4 cell markers *Lrig1* [8] and *Hopx* [10] and found a significantly reduced expression of the former in *Mex3a*-deleted crypts (Fig 2J and K), indicating that there is no compensatory mechanism enforced by the reserve stem cell population in this context. These results were further validated by quantitative real-time PCR (qPCR) expression analysis of the different stem cell markers using isolated crypt fractions from *Mex3a* KO and WT mice (Fig 2L).

In order to directly examine the effect of *Mex3a* deletion on the *Lgr5*⁺ ISC population, we crossed *Mex3a*⁺/⁻ heterozygous mice with the *Lgr5-EGFP-IRES-CreERT2* knock-in mouse model (hereinafter referred to as *Lgr5*⁺/ᴱᴳᶠᴾ), in which EGFP is under the transcriptional control of the endogenous *Lgr5* promoter [2]. *Mex3a*⁺/⁻;*Lgr5*⁺/ᴱᴳᶠᴾ compound mice were then crossed with single *Mex3a*⁺/⁻ heterozygous and *Mex3a*⁻/⁻;*Lgr5*⁺/ᴱᴳᶠᴾ animals were born with a similar frequency to *Mex3a*⁻/⁻ mice, displaying an identical phenotype (Fig EV3A). Most interestingly, 16% of the *Mex3a*⁺/⁻; *Lgr5*⁺/ᴱᴳᶠᴾ double heterozygous mice displayed postnatal lethality (Fig EV3B) associated with disturbances in the intestinal epithelium, including a strong decrease in the global level of EGFP expression (Fig EV3C). Due to the mosaicism generated by random silencing of the construct in the *Lgr5*⁺/ᴱᴳᶠᴾ mouse model [44], EGFP expression does not estimate the actual *Lgr5* expression level in the crypts. To overcome this technical issue, we also performed ISH against *Olfm4* mRNA in these mice and confirmed the loss of *Lgr5*⁺ ISCs (Fig EV3D). Together, these results show that MEX3A expression is somehow required for maintenance of *Lgr5*⁺ stem cells.

## *Mex3a* deletion leads to delayed intestinal epithelial turnover and organoid maturation

To gain evidence concerning a role for MEX3A in regulating *Lgr5*⁺ stem cell function, we isolated *Mex3a* KO crypts and assessed their potential to form multipotent and self-renewing intestinal organoids in culture media containing the minimal ISC niche factor cocktail epidermal growth factor (EGF), noggin (NOG) and R-spondin1 (RSPO1) [3]. At plating, *Mex3a* KO crypts originated smaller structures with a slower growth rate when compared to the WT setting (Fig 3A, day 2), reflecting the tissue condition from which crypts

were isolated. Surprisingly, the *Mex3a* KO crypts were still competent in generating organoids at the end of the first week of culture (Fig 3A, day 6), and we were able to subsequently maintain them for a period of at least 3 months. Nevertheless, after initial culture stabilization, careful inspection disclosed clearly different growth kinetics of KO organoids when compared to WT controls. At the beginning of each passage, we observed that *Mex3a* KO cells originate a higher number of conspicuous spheroids that persist for about 3 days (Fig 3B), eventually turning into budding organoids (Appendix Fig S4). This is in stark contrast with the reduced proportion of spheroids generated from WT cells that mainly consist of budding organoids already at day 2 of culture. After 4–6 days, both cultures exhibit predominantly budding organoids (Fig 3B). In order to understand the difference between *Mex3a* KO and WT organoids, we assessed the expression level of crypt lineage-specific markers (Fig 3C–F). At day 2 of culture, we observed a significant lower mRNA expression of the CBC markers *Lgr5* and *Axin2* in the *Mex3a* KO spheroid population (Fig 3C and F), but not of the +4 cell markers *Hopx* and *Lrig1* (Fig 3F). We could also confirm the presence of a reduced number of Paneth cells by performing qPCR for *Lyz1* mRNA (Fig 3C) and immunohistochemistry for LYZ1 protein (Fig 3D), respectively. At day 4 of culture, the expression of *Lgr5*, *Axin2* and *Lyz1* tended to level off towards the WT organoid levels (Fig 3E). We found no significant differences in terms of proliferative capacity throughout culture. Interestingly, organoid behaviour and gene expression dynamics were recapitulated upon each passage cycle during the analysed time period. Given that *Mex3a* KO crypt epithelial cells still gave rise to intestinal organoids *ex vivo*, we explored a non-cell-autonomous effect of *Mex3a* deletion in mesenchymal cells that could potentially impact *in vivo* paracrine crosstalk with ISCs, promoting the described phenotypes. No significant differences were detected in the expression levels of *Egf, Nog, Rspo1, Wnt2b* and *Wnt3* in the *Mex3a* KO mesenchymal tissue (Fig EV4A). Moreover, it was possible to generate both *Mex3a* KO and WT organoids with KO mesenchymal cell-derived conditioned media in limiting RSPO1 concentrations [45], without an impact in their subsequent self-renewal ability (Fig EV4B).

To address the functional consequence of the reduction of the ISC niche, we chased 5′-bromo-2-deoxyuridine (BrdU) in epithelial cells 3 days after a single pulse. There was a marked reduction in the number of BrdU$^+$ cells present along the crypt-villus axis in *Mex3a* KO mice, in accordance with the smaller pool of proliferative cells present (Fig 3G). In addition, BrdU$^+$ cells were largely located in lower cell positions, while they had significantly progressed up the villi of WT mice during the same period, indicating a delay in the intestinal epithelial migration rate of the mutant mice. Histological analysis provided further proof of this delay. From late embryogenesis until approximately P12, suckling-type enterocytes populate the villi, easily identified in the most distal part of the small intestine by the presence of conspicuous cytoplasmic lysosomal vacuoles. These allow for optimal absorption and digestion of a high-fat/low-carbohydrate milk diet. From P12 onwards, adult-type enterocytes start to be produced, so that by weaning (around P21) only mature cells are present, adapted to the low-fat/high-carbohydrate solid diet [46]. We observed that *Mex3a* KO mice retained a higher percentage of suckling-type enterocytes in time-points when mature enterocytes had almost entirely replaced them in WT animals (Fig EV5). Transmission electron microscopy (TEM) confirmed the extensive presence of these suckling-type enterocytes in the villi of *Mex3a* KO ileum, showing atypical vacuoles with irregular membranous limits, filled with cellular debris and multi-lamellar structures often in the form of whorls of concentric rings (Fig EV5). Overall, the data imply that MEX3A function in the ISC niche is essential to ensure proper intestinal epithelial turnover *in vivo* and typical organoid maturation *ex vivo*.

### *Mex3a* deletion activates the PPAR pathway at the crypt level

In order to delineate the molecular events leading to loss of *Lgr5*-expressing ISCs in *Mex3a* null animals, we acquired the transcriptional profile of isolated crypts from *Mex3a* KO and WT mice using RNA-sequencing (RNA-seq) technology. For these experiments, KO animals were sacrificed before the peak of phenotypic onset in order to preserve the crypt cellular content to its maximum extent. We identified 725 differentially expressed genes (DEGs) in the *Mex3a* mutant crypts compared to WT controls ($P < 0.01$ and $-1.5 \geq$ fold-change $\geq 1.5$; Dataset EV1), 517 of which were upregulated and 208 were downregulated, with a high degree of similarity within genotypes (Fig 4A). Using the Enrichr bioinformatics database [47], gene ontology (GO) assessment showed that biological processes related with lipid homeostasis were significantly enriched in the upregulated class (Fig 4B), including genes like *Abcg5, Abcg8, Acadl, Apoa1* and *Cd36*. On the other hand, terms associated with regulation of mitosis were top ranked in the downregulated class (Fig 4C), including genes like *Aurka, Ccnd1, Cenpa, Plk1* and *Tuba1a*. Interestingly, KEGG pathway analysis revealed a significant enrichment

**Figure 3. Intestinal epithelial turnover and organoid maturation are delayed in *Mex3a* KO mice.**

A  Quantification of the size (diameter length) of *Mex3a* KO and WT organoid-initiating structures 2 days after crypt isolation (*n* = 3 for each genotype, > 150 structures counted in total). Data are represented in a box-and-whisker plot as mean (middle line) with the minimum and maximum distribution values. Each point depicts one organoid-initiating structure. ****$P < 0.0001$, Student's *t*-test. Phase-contrast microscopy images of intestinal organoids generated by *Mex3a*$^{-/-}$ and WT crypts 6 days after crypt isolation. Scale bars, 100 μm.

B  Representative phase-contrast microscopy images of *Mex3a* KO and WT intestinal organoids at days 2 and 4 of culture after 10 weekly passages. Scale bars, 100 μm.

C  qPCR analysis of crypt lineage marker genes in organoid cultures at day 2. Data are presented as the mean fold-change plus standard error (*n* = 4 for each genotype) for target gene expression in *Mex3a* KO intestinal organoids relative to WT (dashed line). *$P < 0.05$, Student's *t*-test.

D  Immunohistochemistry staining for the Paneth cell marker LYZ1 in *Mex3a* KO and WT intestinal organoid sections at day 2 of culture. Scale bars, 50 μm.

E  qPCR analysis of crypt lineage marker genes in organoid cultures at day 4. Data are presented as the mean fold-change plus standard error (*n* = 4 for each genotype) for target gene expression in *Mex3a* KO intestinal organoids relative to WT (dashed line).

F  mRNA ISH staining for the stem cell markers *Mex3a, Lgr5, Hopx* and *Lrig1* in *Mex3a* KO and WT intestinal organoid sections at day 2 of culture. Scale bars, 50 μm.

G  Representative images of immunofluorescence staining for the proliferation marker KI67 and BrdU incorporation in *Mex3a* KO and WT animals. Mice were injected at P17 and sacrificed at P20 (*n* = 3 for each genotype). Scale bars, 50 μm.

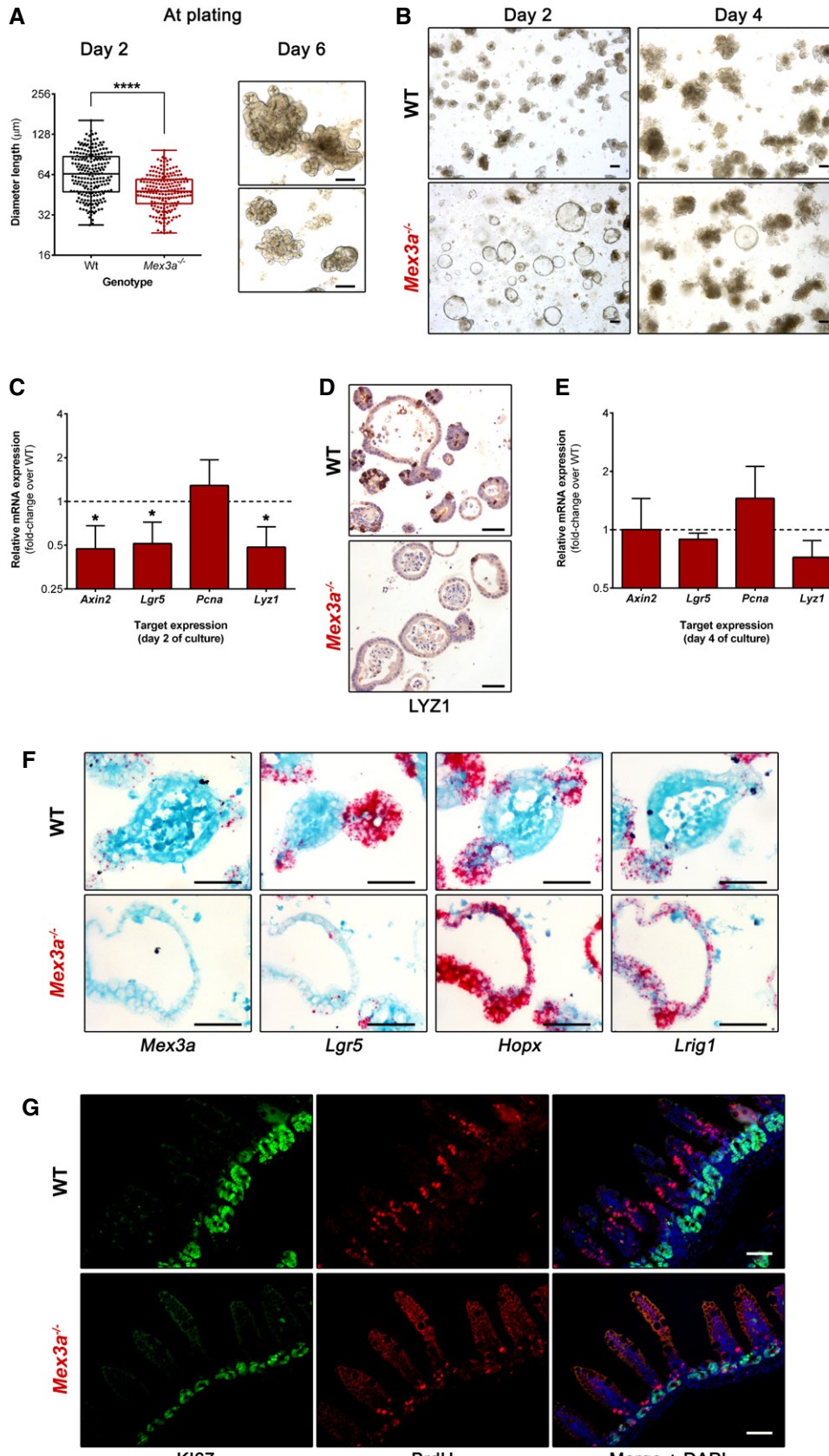

**Figure 3.**

in genes associated with the peroxisome proliferator-activated receptor (PPAR) and the Wnt signalling pathways in the upregulated and downregulated classes, respectively (Fig 4D).

To gather more insights regarding the global pattern of transcriptional changes associated with *Mex3a* deletion, we compared our entire RNA-seq dataset with already defined gene signatures of interest by performing gene set enrichment analysis (GSEA). We found a significant overlap between a recently defined *Mex3a*high gene signature [42] and the downregulated gene class, thus validating our *Mex3a* deletion model (Fig 4E). We also observed an enrichment for the *Lgr5*+ mRNA intestinal stem cell gene signature [40] within the downregulated gene set (Fig 4F).

Furthermore, we detected a significant enrichment for Wnt signalling pathway members [48] within the *Mex3a* KO downregulated gene class (Fig 4G). Finally, we validated by qPCR the induction of selected targets from the PPAR pathway, including *Apoa1*, *Cd36* and *Cyp27a1*, as well as the decreased expression level of different members of the Wnt pathway and the *Lgr5*+ transcriptional signature, including *Ccnd1*, *Fzd2* and *Kcne3* (Fig 4H). Hence, the expression data match our histological findings of a reduced number of *Lgr5*+ ISCs and of proliferative cells upon *Mex3a* deletion, but also point to a putative role of an altered PPAR/Wnt signalling crosstalk associated with the observed phenotypes.

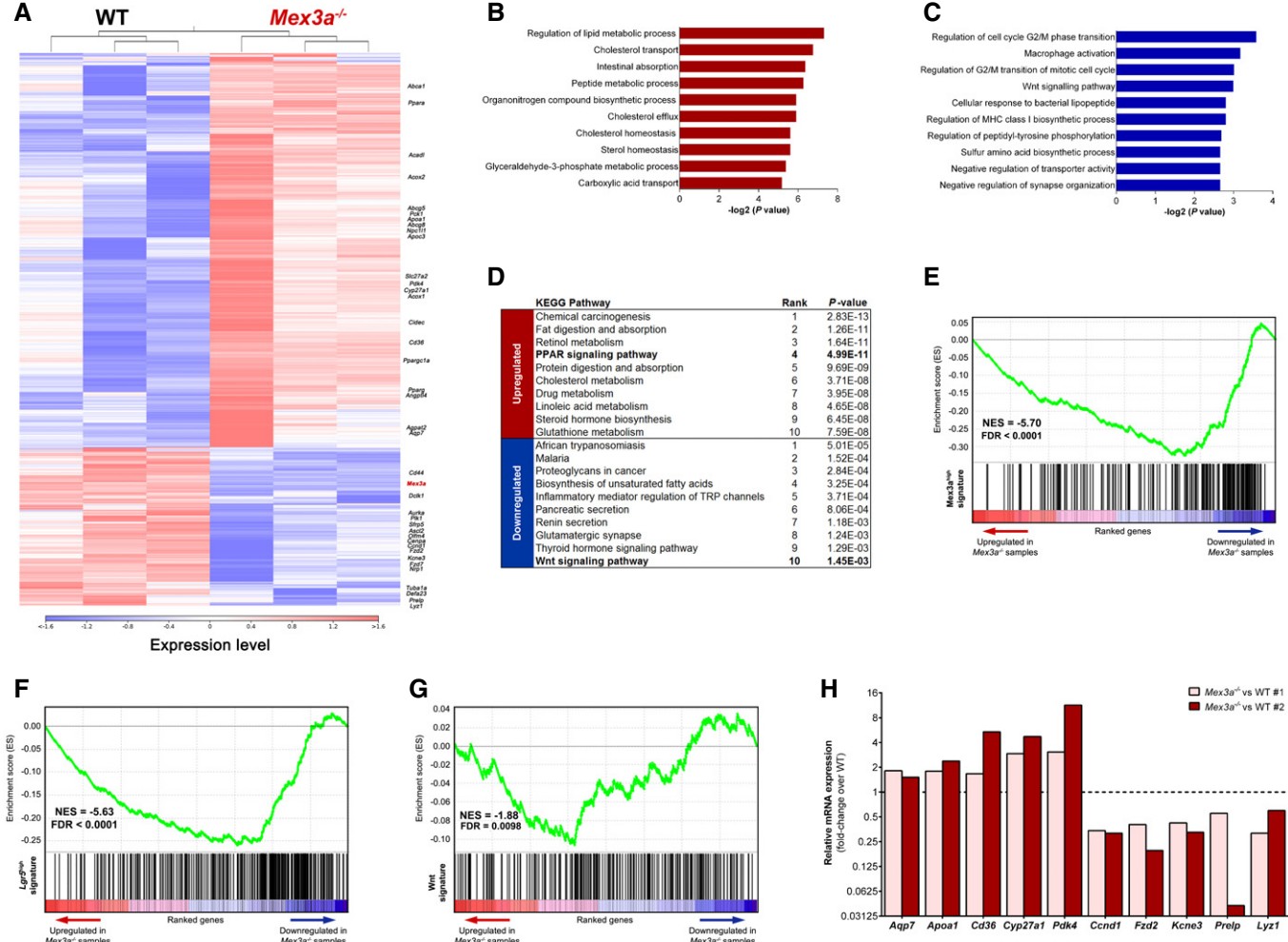

**Figure 4.  *Mex3a* deletion leads to activation of the PPAR pathway at the crypt level.**

A   Hierarchical clustering of samples (*n* = 3 for each genotype) and heatmap of differentially expressed genes between the *Mex3a*−/− and WT mice. The red colour represents overexpression, while the blue colour represents under-expression.

B, C   GO analysis of differentially expressed genes concerning (B) upregulated (red bars) and (C) downregulated (blue bars) biological processes. The 10 most significant terms in each class are depicted according to their *P* value.

D   KEGG pathway analysis of differentially expressed genes. The 10 most significant terms in each class are depicted according to their *P* value.

E–G   GSEA pre-ranked analysis comparing the *Mex3a*−/− transcriptional profile against (E) the *Mex3a*high, (F) the *Lgr5*high and (G) the Wnt signalling pathway gene signatures.

H   qPCR analysis of selected genes in a cohort of independent mice. Data are presented as the mean fold-change of target gene expression in 2 biological replicates (*Mex3a*−/− vs. WT #1 and *Mex3a*−/− vs. WT #2) of *Mex3a* KO intestinal crypts relative to the respective WT controls (dashed line).

## MEX3A regulates PPARγ signalling

The PPAR family of nuclear receptors is comprised of three different isotypes (alpha, beta/delta and gamma) with distinct tissue distribution and ligand specificities. Their activation by endogenous ligands leads to the transactivation of target genes mainly involved in metabolic homeostasis, namely lipid and glucose metabolism [49].

We started by exploring a putative role of MEX3A in the regulation of the PPAR transcription factors by performing transfection experiments in the liver cell line HepG2, which expresses the three PPAR members. Transient MEX3A overexpression led to a 38% decrease in PPARγ protein levels only, without changes in PPARα or PPARβ/δ (Appendix Fig S5A). In order to establish a regulatory link between MEX3A and PPARγ in a more suitable biological context, we used the previously generated Caco-2 intestinal cell line stably transfected with a MEX3A construct [39]. MEX3A overexpression in Caco-2 cells led to a 70% decrease in PPARγ protein levels (Appendix Fig S5B). Moreover, the reduced PPARγ expression was associated with a decrease in AQP7 [50] and CYP27A1 [51] protein levels, two PPARγ transcriptional targets found to be upregulated in the Mex3a null mice (Appendix Fig S5B). Treatment with the specific PPARγ agonist rosiglitazone did not rescue this effect, inducing a slightly higher expression of PPARγ in Caco-2 mock cells only (Appendix Fig S5C). Of note, when we explored the metabolic consequences of MEX3A overexpression, it was apparent that these cells exhibited reduced mitochondrial respiration in both pre-confluent and post-confluent culture conditions compared to Caco-2 mock cells (Appendix Fig S6A and B). Besides, there was a strong inhibition of glycolysis by MEX3A expression in post-confluency (Appendix Fig S6C–F). Thus, in line with the RNA-seq, these data demonstrate that MEX3A expression impacts PPARγ expression and cell metabolism.

Then, we addressed PPARγ expression in mice intestine. We observed a compartmentalized pattern of PPARγ expression in the WT intestine, with increased levels as adult-type enterocytes migrate to occupy the entire villus length (Fig 5A), as previously described [52]. In the Mex3a mutant intestine, this compartmentalization is lost (Fig 5A) and PPARγ is expressed along the entire crypt-villus unit. This different pattern does not seem to be transcriptionally mediated, as Pparg mRNA expression is also detected in the WT crypts (Fig 5B).

To evaluate the putative functional link between increased PPARγ signalling in the crypt and the strong reduction in ISC numbers, we first assessed in intestinal organoids the expression of Pparg and a panel of its target genes upregulated in the RNA-seq data, namely Angptl4 [53], Cidec [54] and Cyp27a1. We observed that the mRNA levels of Pparg and its targets were not significantly different between WT and Mex3a KO-derived organoids (Fig 5C), suggesting that culture media conditions might influence the PPARγ signalling pathway. In a recent study, the removal of glucose or glycolysis inhibition was shown to completely suppress the expansion of hematopoietic progenitor cells promoted by PPARγ antagonism [55]. Thus, to overcome this roadblock and specifically address the effect of PPARγ activity over ISCs, we treated organoid cultures with PPARγ-selective agonists. We observed a dramatic reduction in number and size of both WT and Mex3a KO organoids upon treatment with either rosiglitazone or pioglitazone for 5 days (Fig 5D). Most interestingly, the response was markedly accentuated in Mex3a mutant organoids, as indicated by the decreased level of Lgr5 mRNA expression (Fig 5E) and the lower efficiency of new organoid initiation after passaging when compared to WT organoids (Fig 5F). Overall, combining the RNA-seq and immunohistochemical data in mice with the organoid culture molecular characterization, these results demonstrate that high PPARγ signalling is sufficient to significantly impair Lgr5+ ISC maintenance in the absence of MEX3A, suggesting its increased activity is responsible, at least in part, for the phenotype observed in the Mex3a KO mice.

## Discussion

In this study, we demonstrate for the first time that MEX3A is determinant for maintaining Lgr5+ ISCs and for proper intestinal homeostasis during early postnatal life, in part by controlling PPARγ signalling.

The Mex3a KO mice exhibit a heterogeneous phenotype. On average, only half of the $Mex3a^{-/-}$ mutants are born. About 40% of them survive until adulthood, whereas the remaining 60% exhibit a general failure to thrive and postnatal lethality associated with prominent alterations in the intestinal tract, pinpointing a primary role for MEX3A in intestinal function. In the case of surviving mice, Mex3a functional compensation might occur during embryonic development. This is not surprising, as genes with paralogues are prone to give rise to strains with incomplete penetrance [56]. All four MEX-3 family members share a high degree of homology, particularly at the level of their RNA-binding domains [57]. Therefore, there is a possibility of redundancy in case of ablation of an individual member. We assessed the transcriptional profile of Mex3b, Mex3c and Mex3d in surviving Mex3a KO mice. All members were detected in the small intestine crypts, but no differences were found in mRNA expression levels when compared to WT controls (Pereira B, unpublished observation). It was previously observed by ISH and also using a Mex3a promoter-driven tdTomato reporter mouse that high Mex3a mRNA expression was preferentially located at the +3/+4 crypt position in the adult small intestine [42]. Our analysis showed that during murine postnatal development Mex3a mRNA location is broader, with accumulated expression in the lower portion of the crypts and levelling off towards upper positions. The difference in developmental time-points analysed is probably the main reason behind this discrepancy. ISC hierarchy is not completely defined during postnatal development, without a clear separation between active cycling and quiescent stem cell populations. This is supported by the largely intersecting mRNA expression pattern of both types of ISC markers observed in this work and in other studies during the same period [58,59]. It is possible that Mex3a expression becomes more restricted along time as the stem cell niche becomes fully established.

Given the constitutive nature of the Mex3a KO mouse model here described, it is important to pinpoint the cell of origin of the intestinal phenotype. The restricted expression pattern of Mex3a suggests a major function in the ISC epithelial population. Backing it up, a significant percentage of the $Mex3a^{+/-};Lgr5^{+/EGFP}$ double heterozygous mice also display postnatal lethality associated with decreased Lgr5+ ISC number, akin to the Mex3a single KO mice. It seems Mex3a and Lgr5 combined haploinsufficiency results in a

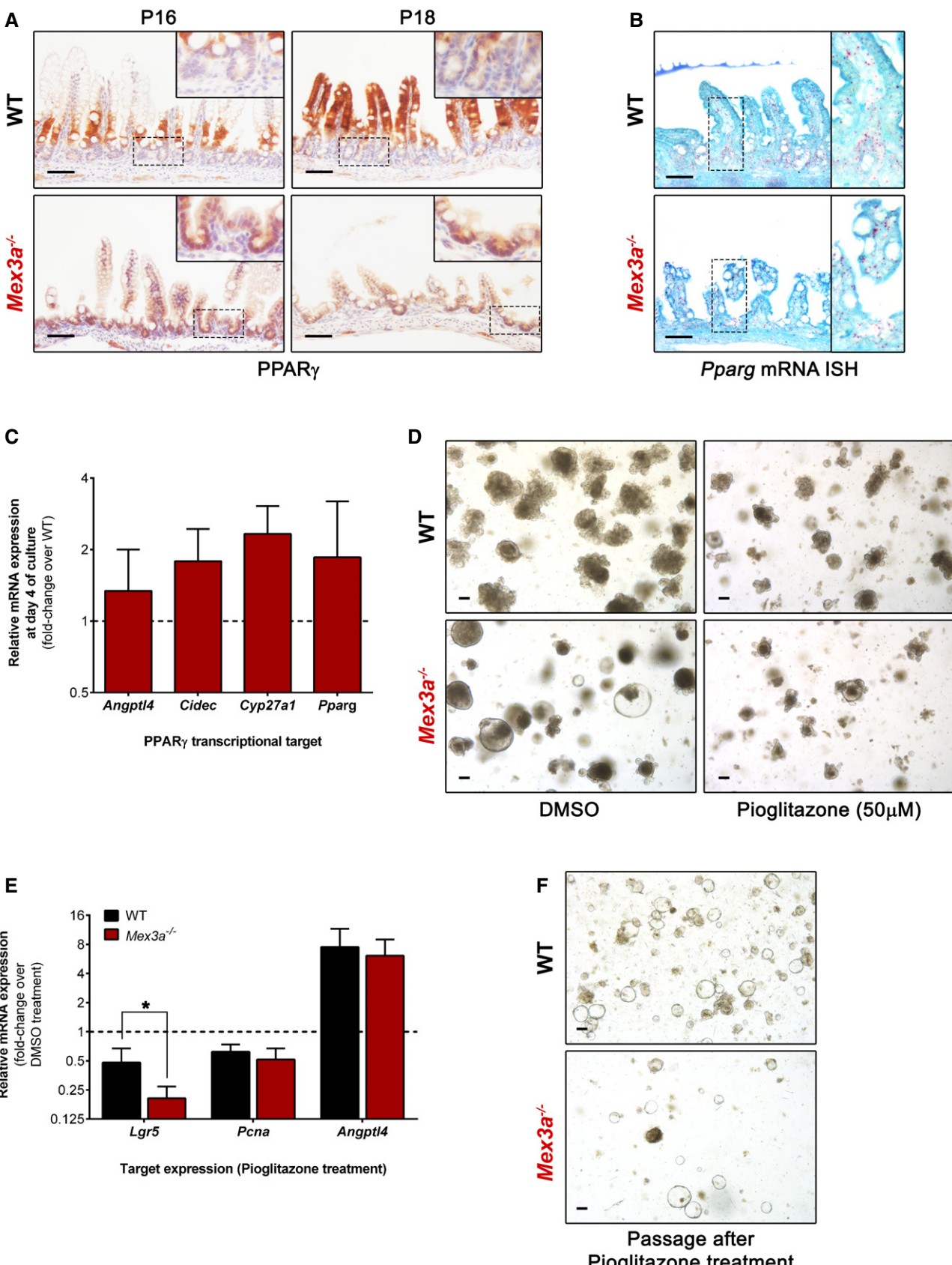

Figure 5.

**Figure 5.  MEX3A regulates PPARγ signalling.**

A   Immunohistochemistry staining for PPARγ in *Mex3a* KO and WT mice ileum. Inserts depict high magnification of the boxed areas. Scale bars, 50 μm.

B   *Pparg* mRNA ISH in ileal sections of *Mex3a* mutants and littermate controls at P18. Inserts depict high magnification of the boxed areas. Scale bars, 50 μm.

C   qPCR analysis of PPARγ transcriptional targets in organoid cultures at day 4. Data are presented as the mean fold-change plus standard error ($n = 3$ for each genotype) for target gene expression in *Mex3a* KO intestinal organoids relative to WT (dashed line).

D   Representative phase-contrast microscopy images of *Mex3a* KO and WT intestinal organoids after 5 days of treatment with 50 μM pioglitazone or vehicle (DMSO). Scale bars, 100 μm.

E   qPCR analysis of *Lgr5*, *Pcna* and *Angptl4* expression in organoid cultures after 5 days of treatment with 50 μM pioglitazone. Data are presented as the mean fold-change plus standard error ($n = 4$ for each genotype) for target gene expression in pioglitazone-treated organoids relative to the DMSO vehicle (dashed line). *$P = 0.02$, Student's *t*-test.

F   Representative phase-contrast microscopy images of *Mex3a* KO and WT intestinal organoids at day 1 after passaging pioglitazone- or vehicle (DMSO)-treated organoids. Scale bars, 100 μm.

cumulative phenotypic effect that can only be confined to ISCs, as this is the only cell population expressing both markers. Furthermore, there is a delayed process of *Mex3a* KO organoid maturation that is recapitulated after each passage, strongly indicating an intrinsic defect to the epithelial ISCs because these are the only ones contributing to organoid formation. Deletion of *Mex3a* in other cell types, such as stromal cells, might also contribute non-cell-autonomously to the observed alterations. However, *Mex3a* mRNA expression was detected at very low levels in the intestinal stromal compartment or not reported to be expressed there [42]. Moreover, we did not find an altered expression profile of genes coding for growth factors secreted from the KO mesenchymal tissue and relevant to ISC maintenance. Lastly, organoids challenged with KO mesenchymal-derived conditioned media maintained their self-renewal potential, suggesting *Mex3a* deletion is unlikely to disturb mesenchymal–epithelial crosstalk. Still, pending on the characterization of a conditional ISC-driven *Mex3a* KO, we cannot formally exclude a contribution from stromal cells or even from a non-intestinal source towards the reported phenotypes.

The striking loss of *Lgr5*⁺ ISCs in *Mex3a* mutant mice could result from either increased apoptosis or altered stem cell identity. We performed immunohistochemistry against activated caspase-3 but did not find increased levels of apoptosis in the *Mex3a* KO mouse crypts (Pereira B, unpublished observation). On the other hand, Paneth cell lineage emergence was severely impaired in the *Mex3a* KO mice during the second and third postnatal weeks, when normally there is an increase in its numbers [60]. Given their status as a component of the stem cell niche by providing crucial Wnt and Notch signals, particularly in the context of *ex vivo* intestinal organoid cultures [61], this feature could explain the decreased *Lgr5*⁺ ISC numbers. Nevertheless, there are also several studies demonstrating that Paneth cells do not play a standalone role in defining the ISC niche. *Lgr5*⁺ cells confined to prospective crypts during the first postnatal days already display stem cell activity, early before the first immature Paneth cells emerge [23]. In agreement, partial or even complete Paneth cell eradication in different mouse models still allows maintenance and proliferation of *Lgr5*⁺ cells [23,62]. More recently, non-epithelial paracrine sources of Wnt ligands have been described that provide a level of redundancy for the niche [63–65]. In this study, isolated *Mex3a* KO crypts originate organoids in the absence of exogenous Wnt supplementation and containing Paneth cells, suggesting that the process of Paneth cell differentiation is not directly hindered by *Mex3a* deletion. Hence, the loss of *Lgr5*⁺ ISCs does not seem to be the consequence of a defect in the Paneth cell lineage. RNA-seq analysis of isolated *Mex3a* KO crypts

indicates an overall downregulation of the *Lgr5*⁺ transcriptional signature as well as of the Wnt signalling pathway, favouring the altered ISC identity hypothesis. The lower number of KI67⁺ proliferative cells, the almost complete absence of the differentiated cell lineages that normally expand postnatally (Paneth and tuft cells), and the delayed epithelial turnover, all point out to an impaired function of the ISC pool. As a final readout, TEM analysis uncovered the abnormal presence of suckling-type enterocytes in the *Mex3a* KO mice during late postnatal stages. These enterocytes are reminiscent of epithelial cells present in patients with malabsorption disorders, having ultrastructural characteristics previously reported as being defective that might compromise nutrient uptake [46]. Ultimately, this could be the cause underlying the observed weight loss and postnatal mortality.

GO and KEGG pathway analyses revealed a significant enrichment for genes associated with PPAR signalling in the *Mex3a* KO crypts. Among the three PPAR family members, PPARβ/δ was reported to have the highest expression level in ISCs, engaging a specific Wnt/β-catenin programme in response to a high-fat diet, leading to increased ISC number and function [66]. This association with high Wnt activity levels contrasts with the changes observed in the *Mex3a* null mice. Besides, PPARβ/δ and PPARα protein levels did not change upon MEX3A overexpression in the cell lines tested, contrary to a MEX3A-mediated downregulation of PPARγ expression. Yet, PPARγ role in the intestinal epithelium and particularly in ISCs has not been tackled. Our data demonstrate that the metabolic output of both *Mex3a* null mice and MEX3A overexpressing cells is altered, possibly mediated by PPARγ function [67]. Contrary to what occurred *in vivo*, PPARγ baseline activity was not different between *Mex3a* KO and WT organoids. Specific microenvironmental cues (or the lack thereof) like growth factor concentration, nutrient availability or circadian rhythms, might allow circumventing the full effect of *Mex3a* deletion in the organoid culture system through distinct ways. For example, allowing maintenance of another stem cell population present in the tissue or de-differentiation of committed progenitors. Such alternative stem cell population might still originate *Lgr5*⁺ cells. In fact, this type of dynamic cellular transitions might be the reason underlying the discrepancies between *in vivo* crypt phenotypes and corresponding *in vitro* organoid cultures observed for different models [12,68]. Because the PPARγ pathway counteracting effect over Wnt signalling is lost *ex vivo*, we used selective agonists to induce PPARγ signalling activation. Overactivation of the PPARγ pathway dramatically impaired organoid formation due to loss of *Lgr5*⁺ ISCs. Observations in other biological settings point to a reciprocally opposing crosstalk with the canonical

Wnt/β-catenin pathway that could be very relevant to explain the *Mex3a* KO mouse phenotype. For instance, PPARγ induction is necessary to suppress Wnt signalling during adipogenesis [69,70]. This regulation also blocks bone marrow progenitors from committing to the osteoblastic lineage [71]. Several *in vitro* studies have shown that PPARγ activation causes growth arrest and differentiation of cancer cell lines, including those derived from lung, prostate, breast, pancreas and colon [72]. Taking into account these considerations, we postulate that the absence of MEX3A *in vivo* results in high PPARγ signalling activity in the crypt, which in turn leads to decreased Wnt/β-catenin pathway and loss of the $Lgr5^+$ transcriptional signature, resulting in ISCs gradual exhaustion and ensuing impairment of all ISC-related functions.

In conclusion, the results presented here provide new insights regarding the functional importance of MEX3A for maintenance of the $Lgr5^+$ stem cell pool and uncover the PPARγ pathway as a novel molecular modulator of stem cell identity. The MEX3A-mediated regulation of PPARγ signalling might also have a yet undisclosed significance for intestinal pathologies, namely cancer initiation/progression.

# Materials and Methods

### Animal models

Animal experimentation was performed in accordance with the Portuguese National Regulation established by Decreto-Lei 113/2013, which is the national transposition of the European Directive 2010/63/EU for the Care and Use of Laboratory Animals. Procedures were evaluated and approved by the i3S Animal Welfare and Ethics Review Body and by the Portuguese National Authority for Animal Health (DGAV)—project licence code no 015434/2017-07-04. The authors involved in executing the procedures (B.P., A.L.A. and N.M.) are certified in animal experimentation (FELASA C course). Mice were bred and maintained at the animal facility of i3S, which is accredited by the Association for Assessment and Accreditation of Laboratory Animal Care (AAALAC), under a standard 12-h light/dark cycle, with water and rodent chow available *ad libitum*. *Mex3a*$^{tm1(KOMP)Vlcg}$ strain was generated at the National Centre of Biotechnology (CNB-CSIC, Madrid, Spain). *Lgr5*$^{tm1(cre/ERT2)Cle}$ and *Meox2*$^{tm1(cre)Sor}$ mice have been previously described and were obtained from The Jackson Laboratory (RRIDs IMSR_JAX:008875 and IMSR_JAX:003755, respectively). The *Meox2*$^{+/Cre}$ mouse model was used as a deleter strain for removal of the floxed neomycin (*Neo*) resistance gene present in the *Mex3a* deletion cassette. To avoid potential problems with the presence of the Cre recombinase, these mice were backcrossed with WT animals to obtain *Mex3a*$^{+/-}$ heterozygous mice (without *Neo* and without *Cre*) from which a colony was established. Animals were genotyped using specific primer pairs (Appendix Table S1) by collecting a minimal portion of tail between P1 and P3. Weights were monitored 3 times a week, and humane endpoints for euthanasia were defined that consisted in gradual weight loss during 3 consecutive weigh-ins, dehydration, lethargy and reluctance to move when stimulated. For assessment of epithelial turnover rate, mice were injected intraperitoneally with 5-bromo-2′-deoxyuridine (BrdU, 10 µl/g, Sigma-Aldrich) 72 h before sacrifice. In general, at least three mice were used for each experiment with corresponding littermates or age-matched animals as controls (detailed description and number of mice are reported in the respective figure legends).

### mRNA *in situ* hybridization (ISH) assay

Manual single-plex ISH was performed using the RNAscope technology (Advanced Cell Diagnostics), whose design ensures selective amplification of target-specific signals and improved signal-to-noise ratios when compared to classic ISH [73]. The RNAscope 2.5 HD Detection Kit-RED was used on formalin-fixed paraffin-embedded (FFPE) mouse tissue and intestinal organoid sections (5 µm) according to the manufacturer's recommendations with the following modifications: (i) after deparaffinization and rehydration, sections were subjected to the mild pre-treatment protocol; (ii) AMP5 incubation was performed for 10 min for *Olfm4* probe (Mm-Olfm4, ref. 311831), 30 min for *Lgr5* probe (Mm-Lgr5, ref. 312171), 30 min for *Lrig1* probe (Mm-Lrig1, ref. 310521), 45 min for *Hopx* probe (Mm-Hopx, ref. 405161), 45 min for *Mex3a* probe (Mm-Mex3a-E2-CDS, ref. 318551) and 1 h for *Pparg* probe (Mm-Pparg, ref. 418821); and (iii) sections were counterstained with haematoxylin in the case of *Olfm4* ISH and fast green stain solution (Thermo Fisher Scientific) in the case of all other probes, for 1 minute. All incubations at 40°C were performed in a HybEZ hybridization oven (Advanced Cell Diagnostics). Slides were dried at 60°C for 15 min and permanently mounted with Vectamount (Vector Laboratories).

### Histochemical procedures

In all procedures, FFPE mouse tissue or intestinal organoid sections (3 µm) were deparaffinized and rehydrated according to standard protocols. For morphological analysis, slides were stained with 7211 haematoxylin for 1 min, differentiated with 1.2% ammonia solution for 30 s and counterstained with eosin Y for 3 min. For the alcian blue–periodic acid–Schiff (AB-PAS) reaction, slides were stained with 1% alcian blue (pH 2.5) for 45 min, treated with 0.5% periodic acid for 10 min and stained with Schiff's reagent for 15 min. Slides were washed and rinsed in deionized after each incubation described above. The nuclei were counterstained with modified Mayer's haematoxylin for 2 min. For immunohistochemistry, heat-induced antigen retrieval was performed in a steamer (Black & Decker) using a 10 mM citrate buffer (pH 6.0) unmasking solution for the majority of antigens for 40 min followed by 20 min cooling at room temperature. In the case of lysozyme, epitope retrieval was performed by enzymatic digestion with 0.04 mg/ml proteinase K in 0.05 M Tris–HCl solution (pH 7.5–7.7) for 5 min at 37°C. Endogenous peroxidase activity was blocked with an aqueous solution of 3% hydrogen peroxide for 10 min. Slides were incubated with normal serum (1:5 in antibody diluent) for 30 minutes before adding the primary antibodies overnight at 4°C (Appendix Table S2). Sections were then incubated with the appropriate biotinylated secondary antibodies for 30 min followed by avidin/biotin complex (Vectastain ABC kit, Vector Laboratories) formation for 30 min according to the manufacturer's protocol. Slides were developed with 3,3′-diaminobenzidine (DAB) chromogen and counterstained with modified Mayer's haematoxylin for 2 min. All slides were dehydrated, clarified and permanently mounted with a xylene-based mounting medium. For

immunofluorescence, after epitope retrieval with citrate buffer, blocking with normal serum (1:5 in antibody diluent) and overnight incubation with primary antibodies at 4°C, sections were incubated with appropriate Alexa Fluor-conjugated secondary antibodies for 30 min, followed by nuclei staining with 1 μl/μg DAPI for 15 min, and mounted with VECTASHIELD Hardset Mounting Medium.

### Transmission electron microscopy

A distal segment of the ileum was flushed with PBS, opened longitudinally and immersed overnight in a fixative solution containing 2.5% glutaraldehyde plus 2% paraformaldehyde in sodium cacodylate buffer 0.1 M (pH 7.4). Samples were washed in 0.1 M sodium cacodylate buffer, post-fixed with 2% osmium tetroxide in 0.1 M sodium cacodylate buffer overnight and incubated in 1% uranyl acetate overnight to improve membrane contrast. Dehydration was performed in a gradient series of ethanol solutions and propylene oxide. Samples were included in EPON resin by immersion in an increasing series of propylene oxide and EPON (until 0:1 ratio) for 60 min each. Sections with 60 nm thickness were prepared on an RMC Ultramicrotome (PowerTome) using a diamond knife and recovered to 200 mesh Formvar Ni-grids, followed by 2% uranyl acetate and saturated lead citrate solution. Visualization was performed at 80 kV in a JEOL JEM 1400 microscope (Jeol), and digital images were acquired using a CCD digital camera Orious 1100W. From each animal, around 20 non-overlapping photomicrographs were obtained. At least three animals per genotype were examined to obtain representative images.

### Crypt isolation

Mouse small intestines were dissected and flushed with ice-cold PBS using a 10-ml syringe mounted with a blunt intestinal probe to remove faeces. The intestine was opened lengthwise and washed three times with ice-cold PBS using vigorous shaking to remove remaining contents. Villi were removed by gently scraping the mucosal side of the tissue with a coverslip, exposing the crypts underneath. The tissue was then cut into pieces $\leq 5$ mm$^2$ with a razor blade, and fragments were incubated in 10 ml of crypt chelating buffer (2 mM EDTA pH 8.0 in PBS) for 1 h at 4°C with gentle rotation. After incubation, the fragments settled, the supernatant was discarded, and 8 ml of ice-cold PBS was added. Fragments were shaken by hand at 3 cycles/s for 1 min to dissociate the epithelium from the basement membrane. At this point, the cell suspension was observed under a microscope to check for crypt enrichment and filtered through a 70 μm mesh into a 50-ml Falcon tube on ice to collect the first crypt fraction. Previous steps were repeated to release remaining crypts still attached to the tissue, combining both fractions in the same tube. The crypt suspension was centrifuged for 10 min at 300 *g* at 4°C. The pellet was suspended in 5 ml ice-cold PBS, and the number of crypts per 10 μl drop was counted.

### Mesenchymal cell isolation and culture

After crypt isolation, intestinal tissue fragments were extensively washed with PBS for three times to remove traces of EDTA and most of the remaining epithelial cells. Thereafter, tissues were digested for 1 h at 37°C in a shaking water bath (200 rpm) in

Dulbecco's modified Eagle's medium (DMEM, Thermo Fisher Scientific) containing 100 μg/ml primocin (InvivoGen), 0.1 mg/ml Liberase TL (Roche Applied Science), 0.04 mg/ml DNase I and 10 μM Y-27632 ROCK inhibitor (both from Stem Cell Technologies). During the digestion step, tissues were further dissociated by vigorous pipetting every 15 min. In the end, samples were filtered through a 70-μm cell strainer, centrifuged at 300 *g* for 5 min, resuspended in DMEM supplemented with 10% FBS and 100 μg/ml primocin and maintained in culture at a 1:2 split ratio for a maximum of three passages.

### Intestinal organoid culture and treatments

Mouse organoids were established and maintained from isolated crypts of the small intestine as described elsewhere [3]. Briefly, around 200 crypts were mixed with Matrigel growth factor reduced (Amsbio) and 40 μl of crypt suspension was placed in the centre of each well on a pre-warmed 24-well plate. After 20 min at 37°C, the Matrigel dome was overlaid with 500 μl of intestinal organoid complete medium. Medium was replaced every 2–3 days, and organoids were passaged every week. The intestinal organoid complete medium was composed of advanced DMEM/Ham's F-12 (Adv. DMEM/F12) with 10 mM HEPES, 1× GlutaMAX, 1 × B-27, 1× N-2 (all from Thermo Fisher Scientific) and 100 μg/ml primocin (InvivoGen). This basic culture medium was supplemented with 50 ng/ml murine recombinant EGF (Peprotech), 10% final volume RSPO1 conditioned medium, 10% final volume NOG conditioned medium (ENR media) and 10 μM Y-27632 (at crypt plating only). Conditioned media were produced using HEK293T cells stably transfected with HA-mouse RSPO1-Fc (a gift from Calvin Kuo, Stanford University) or stably transfected with a mouse NOG-Fc expression vector. For that, Adv. DMEM/F12 supplemented with 10% FBS, 10 mM HEPES and 1× GlutaMAX was conditioned for 1 week with each cell line, filtered through a 0.22-μm syringe filter, and individual working aliquots maintained at −20°C. KO and WT mesenchymal cell cultures were grown at passage 3 in the previous conditions to obtain mesenchymal cell-derived conditioned medium. To address the effect of stromal secreted growth factors on organoid culture dynamics, organoids were cultured with 25% final volume WT or KO mesenchymal cell-derived conditioned media in RSPO1-reduced ENR medium (1% instead of the regular 10%). For ligand stimulation experiments, organoids were treated with vehicle (DMSO) or PPARγ-selective agonists: 50 μM pioglitazone (Sigma-Aldrich) and 50 μM rosiglitazone (Tocris Bioscience) for 5 consecutive days.

### RNA isolation and quantitative real-time PCR

Total RNA was extracted using TRI Reagent according to the manufacturer's guidelines (Sigma-Aldrich). RNA was quantified using a NanoDrop and reverse-transcribed using the Superscript III Reverse Transcriptase Kit (Life Technologies). Analysis of mRNA expression was performed in an ABI Prism 7500 system using the Power SYBR Green Master Mix (Life Technologies) and specific primer pairs (Appendix Table S1). Each sample was amplified in triplicate and specificity confirmed by dissociation analysis. Gene expression was calculated through the relative quantification standard curve method, with *18S* rRNA levels measured for normalization of target gene abundance.

## RNA-sequencing

Three pairs of WT and *Mex3a* null mice (P16) were sacrificed before the peak of phenotypic onset. Total RNA extraction using TRI Reagent was performed as described. Quality control and quantification of total RNA were assessed using the NanoDrop ND-1000 (Thermo Fisher Scientific) and 2200 TapeStation (Agilent Technologies) systems, and only samples with an RNA integrity number above seven were considered for further study. The cDNA library construction was carried out using a Stranded mRNA Library Preparation Kit (Roche Applied Science). The generated DNA fragments were sequenced in the Illumina NovaSeq platform (Illumina), using 150-bp paired-end sequencing reads.

## Bioinformatic analysis

The analysis of the generated sequence raw data was carried out using CLC Genomics Workbench 12.0.3 (Qiagen). Bioinformatic analysis started with trimming of raw sequences to generate high-quality data. For each original read, the regions of the sequence to be removed were determined independently for each type of trimming operation based on the following parameters: (i) Ambiguous limit—2 nucleotides; (ii) Quality limit—0.01 (error probability); (iii) Minimum number of nucleotides in reads—15 nucleotides; and (iv) Discard short reads. The high-quality sequencing reads were mapped against the reference genome *Mus musculus* (GRCm38), using length and similarity fractions of 0.8. The gene expression levels were calculated with the transcripts per million (TPM) method. For differential expression analysis, the differential expression for RNA-seq tool was employed. This multi-factorial statistical analysis tool based on a negative binomial model uses a generalized linear model approach influenced by the multi-factorial EdgeR method, which corrects for differences in library size between the samples and the effects of confounding factors. Genes that presented a fold-change ≥ 1.5 or ≤ −1.5 and a $P < 0.01$ were considered as DEGs. Hierarchical clustering of samples and gene heatmap were generated using the Manhattan distance and average linkage metrics. GO and KEGG pathway analysis was performed using the Enrichr bioinformatics database [47]. GSEA was performed using the freely available software developed by the Broad Institute [74]. The *Mex3a*$^{−/−}$ vs. WT full gene list was ranked according to the log$_2$ fold-change expression values and probed against the indicated gene signatures using the GSEA pre-ranked mode with the following main parameters selected: 10,000 permutations, classic scoring scheme and meandiv normalization.

## Cell culture, transfection and treatments

Human colorectal carcinoma Caco-2 stable cell lines (mock-transfected and overexpressing MEX3A) were previously established [39]. The human liver hepatocellular carcinoma cell line HepG2 is commercially available (American Type Culture Collection, ATCC). These were cultured under standard conditions at 37°C and 5% CO$_2$ in DMEM containing 10% foetal bovine serum, 100 U/ml penicillin and 100 μg/ml streptomycin (Thermo Fisher Scientific). HepG2 transient transfection was performed using the pCMV-MEX3A expression vector and a pCMV-Tag3B empty vector in a ratio of 1 μg DNA to 1.5 μl of Lipofectamine 2000 reagent in OPTI-MEM medium according to the manufacturer's guidelines (Thermo Fisher

Scientific). For assessment of PPAR proteins expression, $2 \times 10^5$ Caco-2 cells were plated per well in a 6-well plate and maintained in culture for 72 h to reach confluence, after which they were collected. For treatment with the PPARγ agonist rosiglitazone (Sigma-Aldrich), $2 \times 10^5$ Caco-2 cells were plated per well in a 6-well plate, allowed to adhere for 48 h and treated with 20 μM rosiglitazone (Tocris Bioscience) or vehicle-treated with DMSO for 24 h, after which they were collected.

## Protein extraction and Western blot analysis

Cells were lysed for 30 min on ice in a lysis buffer containing 20 mM Tris–HCl (pH 7.5), 150 mM NaCl, 2 mM EDTA and 1% IGEPAL (Sigma-Aldrich), supplemented with complete protease inhibitor cocktail (Roche Applied Science), 1 mM PMSF and 1 mM Na$_3$VO$_4$. Lysates were centrifuged at 16,000 *g* for 20 min at 4°C and the supernatant recovered. Protein concentration was determined using the BCA Protein Assay Reagent (Thermo Fisher Scientific). Protein extracts (20–50 μg) were run on 10% SDS–PAGE, transferred to a nitrocellulose membrane and blotted overnight with appropriate antibodies (Appendix Table S2), and signals revealed with ECL detection kit (GE Healthcare Life Sciences). Actin levels were used to normalize protein expression, and quantification was performed using Fiji software [75].

## Seahorse analyses

The extracellular acidification rate (ECAR) and oxygen consumption rate (OCR) were analysed by using the Seahorse XF-Analyzer (Seahorse Bioscience), and $3 \times 10^4$ cells (in pre-confluency) or $2.5 \times 10^5$ cells (in post-confluency) were seeded in 96-well Seahorse plates coated with fibronectin, in normal cell culture medium. The plate was centrifuged at 200 *g* for 1 min. After 30 min of incubation at 37°C, the medium was replaced with Seahorse assay medium specific for OCR or ECAR measurements and equilibrated at 37°C for 1 h in a non-CO$_2$ incubator. OCR and ECAR were measured according to the manufacturer's instructions at baseline and following addition of reagents for the indicated times. Concentrations of reagents were as follows: Oligomycin, 1.5 μM; FCCP, $2 \times 0.5$ μM; Antimycin A and Rotenone, 1 μM; Glucose, 10 mM; 2-DG, 50 μM. For the mitochondrial respiration, the spare respiratory capacity was calculated by subtracting the maximal respiration values by the basal respiration values.

## Statistical analysis

Each experiment was carried out at least three times. Statistical analysis was performed using GraphPad Prism 6.01 software. Differences between groups were considered significant at a $P < 0.05$.

# Data availability

The RNA-sequencing data from this publication have been deposited in the Gene Expression Omnibus database (https://www.ncbi.nlm.nih.gov/geo/) and assigned the identifier GSE141191.

**Expanded View** for this article is available online.

## Acknowledgements

We would like to thank Professor Calvin Kuo (Lokey Center for Stem Biology and Regenerative Medicine, Stanford, USA) for providing us with the HEK293T stably expressing murine RSPO1-Fc cell line and Tiago Duarte (i3S, Portugal) for providing us with the HepG2 cell line. We thank Professor Sobrinho Simões for assistance in electron microscopy image interpretation of cellular ultrastructural characteristics. The authors acknowledge the support of the i3S Scientific Platform Histology and Electron Microscopy (HEMS), member of the national infrastructure PPBI—Portuguese Platform of Bioimaging (PPBI-POCI-01-0145-FEDER-022122). We also thank Sofia Lamas and the animal house technical staff for assistance with the animal models. Part of this work has been supported by the INFRAFRONTIER-I3 project under the EU contract Grant Agreement Number 312325 of the EC FP7 Capacities Specific Programme. This work was financed by the project NORTE-01-0145-FEDER-000029 and supported by Norte Portugal Regional Programme (NORTE 2020), under the PORTUGAL 2020 Partnership Agreement, through the European Regional Development Fund (ERDF). This work was also financed by Fundo Europeu de Desenvolvimento Regional (FEDER) funds through the COMPETE 2020—Operacional Programme for Competitiveness and Internationalisation (POCI), Portugal 2020, and by Portuguese funds through Fundação para a Ciência e a Tecnologia (FCT)/Ministério da Ciência, Tecnologia e Inovação in the framework of the project "Institute for Research and Innovation in Health Sciences" (POCI-01-0145-FEDER-007274) and through the research project POCI-01-0145-FEDER-031538. BP acknowledges FCT for financial support (SFRH/BPD/109794/2015 and CEECIND/03235/2017).

## Author contributions

Conceptualization: BP and RA; Methodology: BP, VMu, GRB, CT, MB and RA; Investigation: BP, ALA, AD, NM, VMu, ARS, CT, AGR and VMá; Writing – Original Draft: BP and RA; Visualization: BP, LD and RA; Writing – Review and Editing: all authors; and Funding acquisition: BP, LD and RA.

## Conflict of interest

The authors declare that they have no conflict of interest.

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
