## [Review Process File · EMBO Reports]

MEX3A regulates *Lgr5*⁺ stem cell maintenance in the developing intestinal epithelium

Bruno Pereira, Ana L. Amaral, Alexandre Dias, Nuno Mendes, Vanesa Muncan, Ana R. Silva, Chantal Thibert, Anca G. Radu, Leonor David, Valdemar Máximo, Gijis R. van den Brink, Marc Billaud, Raquel Almeida

Review timeline:	Submission date:	24 July 2019
	Editorial Decision:	26 July 2019
	Revision received:	18 November 2019
	Editorial Decision:	13 January 2020
	Revision received:	15 January 2020
	Accepted:	20 January 2020

Editor: Achim Breiling

Transaction Report: This manuscript was transferred to *EMBO reports* following peer review at *The EMBO Journal*.

1st Editorial Decision

26 July 2019

Thank you for transferring your manuscript to EMBO reports. I now went through your manuscript, the referee reports from The EMBO Journal (attached below), and your preliminary point-by-point response (revision plan). All referees acknowledge the potential interest of the findings. Nevertheless, they have raised a number of concerns and suggestions to improve the manuscript, or to strengthen the data and the conclusions drawn.

EMBO reports emphasizes novel functional over detailed mechanistic insight, but asks for strong in vivo relevance of the findings, and clear experimental support of the major conclusions. Thus, points regarding further mechanistic insight need not to be addressed; other major concerns and all technical points need to be addressed, though. However, we would not require the conditional KO, and we would not reject the paper, if the organoid transplantation experiments do not work (and you only show the data with conditioned medium). Taken together, the revisions as suggested by your preliminary p-b-p-response seem very reasonable.

Given the constructive referee comments, we would like to invite you to revise your manuscript for EMBO reports with the understanding that the referee concerns must be addressed in the revised manuscript (as indicated above) and/or in a detailed point-by-point response. Acceptance of your manuscript will depend on a positive outcome of a second round of review (using the same referees that have assessed the study before). It is our policy to allow a single round of revision only and acceptance or rejection of the manuscript will therefore depend on the completeness of your responses included in the next, final version of the manuscript.

Revised manuscripts should be submitted within three months of a request for revision; they will otherwise be treated as new submissions. Please contact us if a 3-months time frame is not sufficient for the revisions so that we can discuss the revisions further.

When submitting your revised manuscript, please also carefully review the instructions that follow below. Failure to include requested items will delay the evaluation of your revision. When submitting your revised manuscript, we will require:

- 1) a .docx formatted version of the final manuscript text (including legends for main figures, EV figures and tables), but without the figures included. Please make sure that the changes are highlighted to be clearly visible. Figure legends should be compiled at the end of the manuscript text.
- 2) individual production quality figure files as .eps, .tif, .jpg (one file per figure), of main figures and EV figures. Please upload these as separate, individual files upon re-submission.

The Expanded View format, which will be displayed in the main HTML of the paper in a collapsible format, has replaced the Supplementary information. You can submit up to 5 images as Expanded View. Please follow the nomenclature Figure EV1, Figure EV2 etc. The figure legend for these should be included in the main manuscript document file in a section called Expanded View Figure Legends after the main Figure Legends section. Additional Supplementary material should be supplied as a single pdf labeled Appendix. The Appendix should have page numbers and needs to include a table of content on the first page (with page numbers) and legends for all content. Please follow the nomenclature Appendix Figure Sx, Appendix Table Sx etc. throughout the text, and also label the figures and tables according to this nomenclature.

Important: All materials and methods should be included in the main manuscript file.

For more details please refer to our guide to authors:

See also our guide for figure preparation:

http://wol-prod-cdn.literatumonline.com/pb-assets/embopress/EMBOPress_Figure_Guidelines_061115-1561436025777.pdf

- 4) a complete author checklist, which you can download from our author guidelines (<https://www.embopress.org/page/journal/14693178/authorguide>). Please insert page numbers in the checklist to indicate where the requested information can be found in the manuscript. The completed author checklist will also be part of the RPF.

Please also follow our guidelines for the use of living organisms, and the respective reporting guidelines: <http://www.embopress.org/page/journal/14693178/authorguide#livingorganisms>

- 5) that primary datasets produced in this study (e.g. RNA-seq. data) are deposited in an appropriate public database. See: <http://embor.embopress.org/authorguide#datadeposition>

The accession numbers and database should be listed in a formal "Data Availability" section (placed after Materials & Methods) that follows the model below. Please note that the Data Availability Section is restricted to new primary data that are part of this study.

Data availability

- RNA-Seq data: Gene Expression Omnibus GSE46843
(<https://www.ncbi.nlm.nih.gov/geo/query/acc.cgi?acc=GSE46843>)

- [data type]: [name of the resource] [accession number/identifier/doi] ([URL or identifiers.org/DATABASE:ACCESSION])

6) We strongly encourage the publication of original source data with the aim of making primary data more accessible and transparent to the reader. The source data will be published in a separate source data file online along with the accepted manuscript and will be linked to the relevant figure. If you would like to use this opportunity, please submit the source data (for example scans of entire gels or blots, data points of graphs in an excel sheet, additional images, etc.) of your key experiments together with the revised manuscript. If you want to provide source data, please include size markers for scans of entire gels, label the scans with figure and panel number, and send one PDF file per figure.

7) Our journal also encourages inclusion of *data citations in the reference list* to directly cite datasets that were re-used and obtained from public databases. Data citations in the article text are distinct from normal bibliographical citations and should directly link to the database records from which the data can be accessed. In the main text, data citations are formatted as follows: "Data ref: Smith et al, 2001" or "Data ref: NCBI Sequence Read Archive PRJNA342805, 2017". In the Reference list, data citations must be labeled with "[DATASET]". A data reference must provide the database name, accession number/identifiers and a resolvable link to the landing page from which the data can be accessed at the end of the reference. Further instructions are available at: <http://www.embopress.org/page/journal/14693178/authorguide#referencesformat>

8) Regarding data quantification and statistics, can you please specify, where applicable, the number "n" for how many independent experiments (biological replicates) were performed, the bars and error bars (e.g. SEM, SD) and the test used to calculate p-values in the respective figure legends. Please provide statistical testing where applicable, and also add a paragraph detailing this to the methods section. See: <http://www.embopress.org/page/journal/14693178/authorguide#statisticalanalysis>

9) Please format the references according to our journal style. See: <http://www.embopress.org/page/journal/14693178/authorguide#referencesformat>

I look forward to seeing a revised version of your manuscript when it is ready. Please let me know if you have questions or comments regarding the revision.

REFEREE REPORTS

Referee #1:

The authors of this interesting manuscript report a negative impact on Lgr5+ ISCs after constitutive Mex3a knockout. This study builds on a previous study by Barriga et al. 2017 by characterizing Mex3a expression in intestinal crypts and showing a role for Mex3a in intestinal epithelium maturation. In addition, they demonstrate that Mex3a null mice exhibit PPAR γ activation in intestinal crypts in vivo, and that overactivation of the pathway in vitro results in impaired organoid formation. The authors provide suggestive results linking MEX3A and PPAR signaling to ISCs postnatal maintenance. However, in order to support their conclusions and definitively show that there is a requirement for Mex3a for ISC development or maintenance, the authors should address the following comments.

Major comments:

1. Figure 1 should be moved to supplementary material or added as a panel to Figure 2. The expression pattern of Mex3a has already been characterized by Barriga et al. Further, the colon panels do not add to the story, since the entire manuscript focuses on the small intestine. Finally, the authors should clarify the difference between Mex3a expression seen in the Barriga et al. paper, i.e.

- in +3-4 ISCs, and the expression they report here at the bottom of the crypts, which as they comment later in the manuscript might be due to the early postnatal stages they are using here.
2. It will be informative to compare the villi length they measured in the already shorter ileum villi to the longer duodenum villi, as the mammalian gut shows an antero-posterior development axis. The authors should also discuss the difference between the proximal and distal intestinal phenotype in *Mex3a*^{-/-} mice.
 3. The authors need to do a more in depth characterization of the differentiated cell types, using specific cell markers for the different cell populations, for example: MUC2, CHGA, DLCK1, sucrase-isomaltase, ALDOB.
 4. Is not clear if the reduction in the number of proliferating cells reported here is due to a loss of these cells or because the crypts are smaller; it is possible that the proportion of proliferating cells are maintained. Proliferating cells should be quantify and corrected by crypt size or quantified using FACS.
 5. The authors state that there is a reduction in the amount of *Olfm4* transcripts after *Mex3a* deletion, but why is it that the reduction of *Olfm4* does not equate to the robust reduction of *Lgr5*? Is it possible that *Mex3a* null intestines are not able to respond to Wnt signaling, leading to reduction of *Lgr5* expression in ISCs, together with a direct impact on Paneth cells that respond to WNT. All of this could result in progressively less ISCs.
 6. The *Olfm4*, *Hopx* and *Lrig1* ISH needs quantification, since the current images do not properly show a convincing reduction of the transcripts.
 7. I strongly suggest that the authors use a conditional knockout approach, which would allow them to definitively support their conclusions, as well as to explore ideas such as what is the difference, if any, in *Mex3a* requirement between the high and low *Mex3a* cells reported by Barriga et al. The authors state in lines 321-322, that: "Together, these results strongly suggest that MEX3A expression is required for maintenance of *Lgr5*⁺ stem cells in vivo." With the current genetic model, the authors cannot make this claim. A key experiment to conclude this would be to conditionally knockout *Mex3a* in *Lgr5*⁺ cells using an *Lgr5*CreER driver.
 8. If *Mex3a* is required for *Lgr5*⁺ ISCs, why is it that in organoids *Lgr5* expression levels are similar to wt organoids at day 4?
 9. In Figure 7A, the author nicely show increased expression of PPAR γ in the crypts of *Mex3a*^{-/-} mice. However, the authors state that in *Mex3a*^{-/-} mice, PPAR γ is expressed along the crypt-villus axis, and from their images, it actually seems that the villus expression is reduced in mutant mice.
 10. One of main conclusions of the paper is that *Lgr5*⁺ ISCs are negatively impacted, in number and function, by the loss of *Mex3a*. To support this latter conclusion about function, the authors should carry out an *Lgr5* lineage tracing experiment in vivo and in vitro.
 11. The authors state in lines 591-593 that: "Furthermore, the results obtained with organoid culture indicate that the process is abnormal with delayed transition from spheroid to budding organoids." The delayed transition from spheroid to organoids is not shown.
 12. The authors should measure Wnt/ β -catenin target genes in vivo to assess the impact of MEX3A loss on ISCs and Paneth cells.
 13. A key point of the manuscript is that *Lgr5*⁺ ISCs are lost because of *Mex3a* genetic loss. However, with the current experiments the authors cannot conclude that, because the experiments presented here rely on a constitutive knockout genetic model. *Mex3a* genetic ablation should be done in specific cell populations, such as the *Lgr5*⁺ ISCs. Furthermore, as MEX3A is an RNA binding protein, it is hard to ignore its role in cells and tissues other than the intestine, which could be impacted by using a constitutive genetic model.

Other comments:

1. Missing relevant literature in the introduction regarding:
 - a. Cell plasticity: Yan et al. 2017; Ishibashi et al. 2018; Tomic et al. 2018
 - b. Postnatal expansion of ISCs: Guiu et al. 2019.
2. Line 227 should read increased, instead of reduced?
3. Figure 5 panel A, diameter length is not a proper metric for organoid size and/or growth.
4. Figure 5 panel D, add label indicating the staining, which in this case is LYZ1.
5. Figure 5 panel G, immunofluorescent images are of low quality and the nuclei are not visible by their counterstain. Please improve these images.
6. Figure 7 panel A, add label indicating the staining, PPAR γ in this case.
7. Figure 7 panel B, it is extremely difficult to conclude anything from these images, since the mRNA signal is extremely low. Please improve these images.
8. I recommend the authors use shorter incubation times for the RNAscope probes, which will

decrease background.

Referee #2

How the small intestine develops postnatally in not well characterized, particularly during the transition period of weaning. Here, the authors show that Mex3a is required for proper intestinal development, particularly the transformation that occurs at the weaning stage. They proposed that Mex3a regulates Lgr5⁺ intestinal stem cells via suppression of PPAR γ . The study thoroughly characterizes the phenotype of Mex3a knockouts, but more data is required to convincingly support the link between the Mex3a knockout phenotype and Lgr5 ISC depletion to establish a functional relationship.

1. How do the authors reconcile the fact that SI development and lineage specification (except Paneth cells) are largely normal up until at least 2 weeks of postnatal life in the presence of near-complete loss of Lgr5⁺ ISCs as suggested by the in situ hybridization (ISH)? If there was a minimal number of functional Lgr5⁺ ISCs (and also other alternative stem cell pools as suggested by reduced Hopx and Lrig1 expression), wouldn't the authors expect a more drastic phenotype earlier?
2. To evaluate the presence of surviving Lgr5⁺ ISCs (perhaps due to some mosaicism from rescue from Mex3a paralogous members), can the authors use lineage tracing with Mex3aKO;Lgr5-EGFP-IRES-CreERT2;R26-conditional reporter (e.g. tdTomato) to show the turnover and what are the cell types that can be generated from the Mex3aKO ISCs?
3. Similar to point #1, with a marked reduction of Lgr5⁺ ISCs, how are the organoids sustainable? There is no mention of the longevity of the organoids derived from Mex3a knockout crypts in culture - Can they be propagated for as long as the WT crypts? Or can they only be propagated for a limited time? This reflects the functionality of Lgr5⁺ ISCs that remain in the organoids.
4. As the mouse model is a systemic constitutive knockout of Mex3a, can the authors perform transplantation of the organoids into a wildtype host to ascertain the ISC-specific phenotype?
5. It is difficult to reconcile the contrasting expression patterns of non-CBC-specific stem cell marks Hopx and Lrig1 in Mex3a knockout SI (in which they are downregulated) and organoids (in which they are upregulated). What do the authors make of these opposing observations?
6. RNASEQ and qPCR validation performed on only 2 mice per group - I disagree with what the authors said about the "high degree of similarity within genotypes" (line 411). Looking at the heatmap in Fig 6A, the 4 samples are visibly different from one another, and the two most similar samples are a Mex3a knockout and WT (two rightmost columns). Between the two Mex3a knockouts, the magnitudes of upregulation in the most highly upregulated genes are don't concur. Therefore, the authors should increase the sample size for the transcriptome profiling and qPCR validation as this has ramifications for the later PPAR γ story.
7. The authors need to show positive control PPIB for the ISH experiments to show that there is no global downregulation of all transcripts since Mex3a is an RNA binding protein. Relating to this point, the RNA in situ hybridization panels should be supported by qPCR data as well since qPCR allows for quantitative normalization of each tissue to a housekeeping gene and would address whether the lower expression of the stem cell markers are due to a global downregulation of gene expression (e.g. Figs 3, 4, 5F).
8. Can the authors perform the complementary experiment to Fig 7 and culture Mex3a knockout organoids in PPAR γ antagonist to see if it can partially rescue the morphological phenotype and increase Lgr5 expression?

Referee #3

The manuscript by Pereira and colleagues describes how Mex3a controls maintenance of Lgr5-positive intestinal stem cells. The authors show that Mex3a is required for proliferation of ISCs as well as for differentiation of Paneth cells. They conclude that Mex3a negatively regulates PPAR γ protein levels in the crypt cells. They further demonstrate that PPAR γ signaling impairs growth of intestinal organoids. The manuscript is well written and of general interest. There is one point that need to be addressed to strengthen the conclusions the authors draw.

Major point:

Mex3a was ablated in both epithelium and mesenchyme. Therefore, it is possible that the absence of Wnt and other signals from the mesenchymal cells are responsible for the proliferation defects observed in the mutant mice. The expression of Wnts, BMP antagonists and EGF could be evaluated by qPCR analysis of the mesenchymal cells.

Minor point:

Lane 466: The authors write: "MEX3A overexpression in Caco-2 cells led to a 70% decrease in PPAR γ protein levels (Figure S6B)." However, based on their quantification, the protein levels are decreased to 70%. These is a 30% decrease only. Here, the authors should formulate the statement clearly.

1st Revision - authors' response

18 November 2019

EMBOR-2019-48938-T**Referee comments:****Referee #1:**

The authors of this interesting manuscript report a negative impact on Lgr5+ ISCs after constitutive Mex3a knockout. This study builds on a previous study by Barriga et al. 2017 by characterizing Mex3a expression in intestinal crypts and showing a role for Mex3a in intestinal epithelium maturation. In addition, they demonstrate that Mex3a null mice exhibit PPAR γ activation in intestinal crypts in vivo, and that overactivation of the pathway in vitro results in impaired organoid formation. The authors provide suggestive results linking MEX3A and PPAR signaling to ISCs postnatal maintenance. However, in order to support their conclusions and definitively show that there is a requirement for Mex3a for ISC development or maintenance, the authors should address the following comments.

Major comments:

1. Figure 1 should be moved to supplementary material or added as a panel to Figure 2. The expression pattern of Mex3a has already been characterized by Barriga et al. Further, the colon panels do not add to the story, since the entire manuscript focuses on the small intestine. Finally, the authors should clarify the difference between Mex3a expression seen in the Barriga et al. paper, i.e. in +3-4 ISCs, and the expression they report here at the bottom of the crypts, which as they comment later in the manuscript might be due to the early postnatal stages they are using here.

We appreciate that the Reviewer considers our study interesting. We agree with the Reviewer's suggestion and have now included the small intestine and colon panels in Fig EV1, which provides for the first time a comprehensive characterization of the *Mex3a* expression pattern in major organs of the mouse. Furthermore, we performed this analysis using the corresponding *Mex3a* KO tissues as the best possible control to confirm the specificity of the *Mex3a* mRNA *in situ* hybridization (ISH) probe. Thus, this characterization goes beyond the study by Barriga et al. 2017. Regarding the distinct *Mex3a* mRNA localization profile in the intestinal crypts between both studies, we hypothesize that it might be the result of the developmental time-point studied and made this point clearer in the Discussion section. In addition, distinct ISH probes were used to detect *Mex3a* (targeting the mRNA 3'UTR in Barriga et al. 2017 or the coding sequence in our study), which might lead to different outcomes as a result of RNA secondary structure or due to distinct occupancy profiles by *trans*-regulators (such as microRNAs or RNA-binding proteins). From a technical standpoint, this might affect the hybridization process, particularly in the case of the 3'UTR probe.

2. It will be informative to compare the villi length they measured in the already shorter ileum villi to the longer duodenum villi, as the mammalian gut shows an antero-posterior development axis. The authors should also discuss the difference between the proximal and distal intestinal phenotype in Mex3a^{-/-} mice.

We thank the Reviewer for pointing this out. We also characterized the proximal small intestinal region of the *Mex3a* KO mice. The phenotype follows the same trend described for the distal part regarding histological alterations, but it is less pronounced. For instance, while a 22% decrease in

crypt depth was observed in the proximal region of the *Mex3a* KO intestine, in the ileum we observed a 41% decrease, justifying our focused analysis in the most distal region. A possible explanation is that there are regional differences in the intestinal epithelial turnover dynamics. It is known that the ileum displays the fastest turnover rate in mice and rats (Leblond and Stevens, 1948; Darwich *et al.* 2014), making it more vulnerable to alterations in ISC numbers. Also, specific ISC markers like *Bmi1* have an expression pattern apparently restricted to the proximal small intestine (Sangiorgi and Capecchi, 2008), providing an additional layer of plasticity that might offer some type of functional compensation in this region. The data on the proximal region is now included in Appendix Fig S2.

3. The authors need to do a more in depth characterization of the differentiated cell types, using specific cell markers for the different cell populations, for example: MUC2, CHGA, DLCK1, sucrase-isomaltase, ALDOB.

We partially disagree with the Reviewer on this point. We have provided a characterization of the main differentiated cell types in Fig EV2, where Sucrose-isomaltase staining (SIS) is already shown. We did not assess MUC2 expression but used alcian-blue periodic acid Schiff (AB-PAS) staining to label goblet cells. This histochemical technique labels mucins and in intestine the most abundant mucin is MUC2 that is produced by goblet cells, so the result is equivalent. Enteroendocrine cells were labelled with synaptophysin (SYP) and enterocytes were labeled with Villin (VIL1), SIS and CDX2. However, we had not assessed the tuft cell lineage using the DCLK1 marker and thank the Reviewer for pointing it out. We now provide this data in Fig 2E. The number of DCLK1+ cells is significantly decreased in the *Mex3a* KO intestine. Tuft cells constitute a rare and unique post-mitotic differentiated intestinal lineage with a chemosensory role. This lineage is derived from *Lgr5*-expressing ISCs and the first tuft cells appear around P7 becoming readily detected around P14 (Gerbe *et al.*, 2011). Because tuft cells present a postnatal emergence like Paneth cells, their decreased number in the *Mex3a* KO mice is also explained by the significant loss of *Lgr5*+ ISCs, thus reinforcing the functional effect of *Mex3a* deletion over normal intestinal epithelial turnover. Of note, the ablation of DCLK1+ cells using the *Rosa26-iDTR* mice is well tolerated under homeostatic conditions, without clinical signs of gastrointestinal pathology (Westphalen *et al.*, 2014).

4. Is not clear if the reduction in the number of proliferating cells reported here is due to a loss of these cells or because the crypts are smaller; it is possible that the proportion of proliferating cells are maintained. Proliferating cells should be quantify and corrected by crypt size or quantified using FACS.

We have now quantified the number of proliferating cells. The average number of KI67+ cells per crypt in *Mex3a* mutant mice is 6.56 ± 1.39 compared with 12.60 ± 1.07 in WT mice, a 48% difference. The significant lower number is a consequence of the loss of ISCs, as these are the ones that would originate the KI67+ TA population, as well as the Paneth and tuft cell lineages, which are almost absent. As requested, we normalized the number of KI67+ cells to crypt size. The proportion is not significantly different, although there is a tendency to be decreased in the *Mex3a* KO mice (0.35 ± 0.07 compared with 0.41 ± 0.01 in WT mice). This is not unexpected because crypt size, without a gain of any of the other populations, is in fact determined by the number of TA cells (the main cellular component of crypts). Due to loss of the *Lgr5*+ ISCs, there is no continuous replenishment of the TA cell population. Additionally, the remaining TA population will differentiate into other epithelial cell lineages (although slowly, as indicated by the BrdU incorporation assays), and so both events contribute to an overall reduction in the TA cell number and, consequently, smaller crypt size. This data is now included in Appendix Fig S3.

5. The authors state that there is a reduction in the amount of Olfm4 transcripts after Mex3a deletion, but why is it that the reduction of Olfm4 does not equate to the robust reduction of Lgr5? Is it possible that Mex3a null intestines are not able to respond to Wnt signaling, leading to reduction of Lgr5 expression in ISCs, together with a direct impact on Paneth cells that respond to WNT. All of this could result in progressively less ISCs.

There is always a robust decrease in both markers in *Mex3a* KO mice (please see also our response to point 6). It is well documented that *Olfm4* presents higher endogenous mRNA levels than *Lgr5* in ISCs (Itzkovitz *et al.* 2012; Schuijers *et al.* 2014; Yan *et al.* 2017). This is the main reason why it is used as a standard marker for visualization of *Lgr5*+ stem cells by ISH (van der Flier *et al.* 2009). Due to this difference in expression profile, *Olfm4* is still noticeable in instances where there are few *Lgr5*+ cells present in the tissue and *Lgr5* expression is already residual (our study and Yan *et al.*

2017). Nevertheless, we have also observed *Mex3a* KO intestinal tissues where *Olfm4* expression is almost completely lost (please see Reviewer Figure 1 below).

Reviewer Figure 1. *Olfm4* mRNA expression detected by ISH in two additional pairs of *Mex3a* KO and WT mice (P16 and P19).

6. *The Olfm4, Hopx and Lrig1 ISH needs quantification, since the current images do not properly show a convincing reduction of the transcripts.*

We have performed quantitative real-time PCR (qPCR) for these markers in isolated crypt fractions. The analysis is now included in Fig 2L and confirms the significant reduction in *Lgr5*, *Olfm4* and *Lrig1* mRNA expression levels observed by ISH. *Hopx* mRNA levels are not significantly different between WT and *Mex3a* KO crypts.

7. *I strongly suggest that the authors use a conditional knockout approach, which would allow them to definitively support their conclusions, as well as to explore ideas such as what is the difference, if any, in Mex3a requirement between the high and low Mex3a cells reported by Barriga et al. The authors state in lines 321-322, that: "Together, these results strongly suggest that MEX3A expression is required for maintenance of Lgr5+ stem cells in vivo." With the current genetic model, the authors cannot make this claim. A key experiment to conclude this would be to conditionally knockout Mex3a in Lgr5+ cells using an Lgr5CreER driver.*

We agree this experiment would reinforce our data on the role of intestinal MEX3A expression in *Lgr5*⁺ cells. Unfortunately, we currently do not have such model and, as the Reviewer might recognize, we would not be able to establish it within a reasonable timeframe for this revision. Therefore, we must restrict our analysis to the genetic model at hand and consider its strengths while acknowledging its limitations. We want to emphasize that we are describing the first *Mex3a* deletion model, which is part of the MEX-3 family of RNA-binding proteins whose biological functions are not well known. The present study goes far beyond the two main existing publications studying MEX3A. One is our previous work, where we described an *in vitro* association between MEX3A, loss of differentiation and increased expression of ISC markers, including *LGR5* (Pereira *et al.* 2013). The other describes *Mex3a* as a marker of slowly proliferating *Lgr5*⁺ cells located around position +3/+4 (Barriga *et al.* 2017) but does not provide direct functional data on the MEX3A role. We now provide evidences, outlined below, strongly supporting a specific role of intestinal epithelial MEX3A expression for the maintenance of *Lgr5*⁺ ISCs:

1. The restricted expression pattern of *Mex3a* in the base of the intestinal crypts suggests a function in the epithelial population, even more so when compared with the very low and dispersed mRNA expression observed in the mesenchymal compartment. In accordance, using a transcriptional reporter for *Mex3a* expression driving a tdTomato protein (*Mex3a*^{Tom/+} reporter mouse), Barriga *et al.* 2017 only showed *Mex3a* expression in a subpopulation of *Lgr5*⁺ ISCs and did not report expression in the mesenchyme. We also did not detect any obvious alteration concerning the underlying mesenchymal morphology of the *Mex3a* KO mice;

2. About 16% of the double *Mex3a*^{+/-};*Lgr5*^{+EGFP} heterozygous mice display an intestinal phenotype similar to the *Mex3a* KO, with loss of *Lgr5*-expressing cells. This is highly significant, as both the *Mex3a*^{+/-} heterozygous mice and the *Lgr5*^{+EGFP} do not present any phenotype on their own. This indicates that *Mex3a* and *Lgr5* combined haploinsufficiency results in a cumulative effect that can only be confined to ISCs, as this is the only cellular population expressing both markers;
3. Although we can maintain KO organoids (indicating there is a population with stem cell features), we observe a substantial delay in the normal timing of intestinal organoid maturation upon each organoid passage, which includes reduced expression of the CBC markers *Lgr5* and *Axin2*, and of the Paneth cell marker *Lyz1*, demonstrating an epithelial cell-intrinsic defect;
4. No significant differences were detected in the KO mesenchymal tissue concerning the expression level of genes coding for specific growth factors (*Egf*, *Nog*, *Rspo1*, *Wnt2b* and *Wnt3*), indicating that *Mex3a* ablation does not alter the expression profile of mesenchymal genes relevant for ISCs maintenance;
5. It is possible to generate and passage both KO and WT organoids in the presence of conditioned media derived from KO stromal cells, indicating that a putative *Mex3a* KO mesenchymal contribution in the culture system does not impact stem cell self-renewal.

8. If Mex3a is required for Lgr5+ ISCs, why is it that in organoids Lgr5 expression levels are similar to wt organoids at day 4?

The *ex vivo* organoid culture is a semi-physiological system that allows maintenance of ISCs in conditions that are different from the actual *in vivo* setting. Specific microenvironmental cues provided by growth factors and/or nutrients present in the enriched culture setting might allow bypassing the full effect of *Mex3a* absence in distinct ways. For example, allowing maintenance of a stem cell population already present in the tissue or de-differentiation of committed progenitors. This alternative stem cell population might originate *Lgr5*⁺ cells. In fact, this might be the reason why functional uncoupling between *in vivo* crypt phenotypes and corresponding *in vitro* organoid cultures has been recurrently observed in other models. For example, *Lgr5*⁺ ISCs are lost from the crypt upon conditional *Ascl2* deletion (van der Flier *et al.* 2009) but are present in the matching organoids (Schuijers *et al.* 2015). Crypts depleted of *Lgr5*-expressing CBCs by treatment with diphtheria toxin (*Lgr5*^{DTR/+} mice) give rise to organoids with similar efficiency as WT controls (Tian *et al.* 2011). Intestinal organoids from *Lgr5*^{-/-} homozygous mice have also been generated and maintained in culture (Carmon *et al.*, 2017). This is an interesting observation and we will continue to pursue it, but a deeper understanding at this stage falls outside the scope of the current work, as it does not argue against the effect of *Mex3a* deletion in ISCs.

9. In Figure 7A, the author nicely show increased expression of PPARγ in the crypts of Mex3a-/- mice. However, the authors state that in Mex3a-/- mice, PPARγ is expressed along the crypt-villus axis, and from their images, it actually seems that the villus expression is reduced in mutant mice.

PPARγ expression is associated with intestinal differentiation, and its expression becomes compartmentalized as adult-type enterocytes migrate to occupy the entire villi (Chen *et al.*, 2006). By looking carefully to Fig 5A, it is perceptible that suckling-type enterocytes at P16 (distinguishable due to the presence of large vacuoles) do not stain for PPARγ. *Mex3a* KO mice show an impaired epithelial turnover rate, and because of that this type of immature enterocytes is still present at P18 when they should already be replaced. This is the reason why the expression in the villi is not as strong as compared to WT controls. Most importantly, PPARγ is present in the *Mex3a* KO crypts.

10. One of main conclusions of the paper is that Lgr5+ ISCs are negatively impacted, in number and function, by the loss of Mex3a. To support this latter conclusion about function, the authors should carry out an Lgr5 lineage tracing experiment in vivo and in vitro.

Unfortunately, we do not have a *Mex3a*;*Lgr5* lineage-tracing ready strain to conduct this experiment within a reasonable time-frame for the revision, as this involves at least two mouse crosses (for instance, *Rosa26*^{F/F} x *Mex3a*^{-/-} and *Rosa26*^{F/+};*Mex3a*^{+/-} x *Mex3a*^{+/-};*Lgr5*^{+EGFP}) and all the necessary husbandry steps between them. Nevertheless, the absence of the two differentiated cell lineages that normally expand postnatally (Paneth and tuft cells), the BrdU pulse-chase experiments and the ultrastructural characterization by electron microscopy of the type of enterocytes present in the villi of *Mex3a* KO mice, all demonstrate that there is an impairment of *Lgr5*⁺ ISCs renewal function upon *Mex3a* loss.

11. *The authors state in lines 591-593 that: "Furthermore, the results obtained with organoid culture indicate that the process is abnormal with delayed transition from spheroid to budding organoids." The delayed transition from spheroid to organoids is not shown.*

We agree with the Reviewer that this transition was not illustrated. We have now included this data in Appendix Fig S4.

12. *The authors should measure Wnt/ β -catenin target genes in vivo to assess the impact of MEX3A loss on ISCs and Paneth cells.*

Both gene ontology and KEGG pathway analysis of our differentially expressed gene list show that the Wnt/ β -catenin signalling pathway is significantly downregulated in the *Mex3a* KO crypts, including genes like *Ascl2*, *Fzd2*, *Fzd7*, *Ccnd1*, *Cd44*, *Kcne3*, and *Sfrp5*, among others. This is also backed up by GSEA analysis of the entire dataset against a list of Wnt target genes (compiled from van der Flier *et al.* 2007 and the KEGG_WNT_SIGNALING_PATHWAY curated gene set available in the Broad Institute Molecular Signatures Database), which shows a significant enrichment for the Wnt signature within the *Mex3a* KO downregulated gene class. In addition to *Fzd2* and *Kcne3*, we have now also validated *Ccnd1* by qPCR in isolated crypt fractions. The data is included in Fig 4H.

13. *A key point of the manuscript is that Lgr5+ ISCs are lost because of Mex3a genetic loss. However, with the current experiments the authors cannot conclude that, because the experiments presented here rely on a constitutive knockout genetic model. Mex3a genetic ablation should be done in specific cell populations, such as the Lgr5+ ISCs. Furthermore, as MEX3A is an RNA binding protein, it is hard to ignore its role in cells and tissues other than the intestine, which could be impacted by using a constitutive genetic model.*

We thank the Reviewer for this analysis. We are aware that we are only characterizing in depth the phenotype in the intestine and might be facing confounding effects due to alterations in other organs. However, the fact that the intestinal phenotype is the only striking alteration detected upon macroscopic observation and histological analysis of major organs, associated with a general failure of the animals to thrive within 3 weeks post-birth indicates that intestinal malfunction is a major cause of death. As stated in the response to point 7, we can make the claim that *Mex3a* genetic ablation leads to loss of *Lgr5*+ ISCs in the intestine because this is the observed outcome through different methodologies. What we cannot state is that this is the consequence of loss of *Mex3a* expression solely in the intestinal epithelial *Lgr5*+ cells. There might be a contribution of loss of expression from other cell types, particularly the mesenchyme, as the Reviewers point out. While we must consider this hypothesis due to the constitutive nature of our genetic model, we do find it very unlikely, based mainly on three evidences we provide: 1) the expression pattern of *Mex3a*, as it is much more expressed in the intestinal crypt than in any other compartment of the intestine; 2) when we combine loss of one *Mex3a* allele with loss of one *Lgr5* allele we get the same phenotype observed with loss of the two *Mex3a* alleles, indicating a specific effect in the *Lgr5*+ cells; 3) we did not detect expression differences in genes coding for growth factors secreted by the *Mex3a* KO mesenchyme tissue *in vivo* nor an impact over stem cell self-renewal in the organoid culture system using KO mesenchymal-derived conditioned media. Still, in agreement with the Reviewers opinions, we have tried to clearly highlight the strengths and weaknesses of our genetic model in the Discussion section of the revised manuscript.

Other comments:

1. *Missing relevant literature in the introduction regarding:*

a. *Cell plasticity: Yan et al. 2017; Ishibashi et al. 2018; Tomic et al. 2018*

b. *Postnatal expansion of ISCs: Guiu et al. 2019.*

We thank the Reviewer for pointing these out. The missing references were added to the revised manuscript.

2. *Line 227 should read increased, instead of reduced?*

We clarified this sentence.

3. *Figure 5 panel A, diameter length is not a proper metric for organoid size and/or growth.*

In this early stage (at day 2 after plating), freshly isolated crypts are essentially cystic/round structures. Hence, diameter length provides a good estimate of the size in this time-point.

4. Figure 5 panel D, add label indicating the staining, which in this case is LYZ1.

The label was added.

5. Figure 5 panel G, immunofluorescent images are of low quality and the nuclei are not visible by their counterstain. Please improve these images.

The resolution of the immunofluorescence images was improved.

6. Figure 7 panel A, add label indicating the staining, PPAR γ in this case.

The label was added.

7. Figure 7 panel B, it is extremely difficult to conclude anything from these images, since the mRNA signal is extremely low. Please improve these images.

The resolution of the ISH images was improved.

8. I recommend the authors use shorter incubation times for the RNAscope probes, which will decrease background.

We thank the Reviewer for the suggestion.

Referee #2:

How the small intestine develops postnatally in not well characterized, particularly during the transition period of weaning. Here, the authors show that Mex3a is required for proper intestinal development, particularly the transformation that occurs at the weaning stage. They proposed that Mex3a regulates Lgr5+ intestinal stem cells via suppression of PPAR γ . The study thoroughly characterizes the phenotype of Mex3a knockouts, but more data is required to convincingly support the link between the Mex3a knockout phenotype and Lgr5 ISC depletion to establish a functional relationship.

1. How do the authors reconcile the fact that SI development and lineage specification (except Paneth cells) are largely normal up until at least 2 weeks of postnatal life in the presence of near-complete loss of Lgr5+ ISCs as suggested by the in situ hybridization (ISH)? If there was a minimal number of functional Lgr5+ ISCs (and also other alternative stem cell pools as suggested by reduced Hopx and Lrig1 expression), wouldn't the authors expect a more drastic phenotype earlier?

We appreciate that the Reviewer recognizes the comprehensive nature of our report on the Mex3a KO phenotype. The existing studies focusing on postnatal intestinal development indicate the following:

1. Mice are born with an immature intestinal epithelium that contains villi but lacks crypts. At this stage, only a small number of Lgr5+ progenitors exist, restricted to basal regions between the villi called intervillus domains (Kim *et al.* 2012; Yanai *et al.* 2017). These LGR5+ cells confined to prospective crypts already display stem cell capacity early before the first immature Paneth cells emerge (only around P7), suggesting Paneth cells are not required in initial stages of postnatal development for the stem cell niche;
2. Fetal LGR5 progeny is, by itself, insufficient to sustain intestinal growth during initial development (Guiu *et al.* 2019). So, intestinal postnatal development might also be dependent on Lgr5- ISC populations. Along the same line, Lgr5+ ISCs seem dispensable for short-term crypt maintenance in different adult models (Tian *et al.* 2011; Yan *et al.* 2012; Yan *et al.* 2017);
3. It is known that villus epithelial cells have a longer life span (10 to 15 days) at the postnatal stage, because the intestinal epithelium is not in a steady state condition as in the adult (Al-Nafussi and Wright, 1982; Cheng and Bjerknes, 1985; Calvert and Pothier, 1990). The crypt population also does not change significantly during the first 10 days of postnatal life (Sumigray *et al.* 2019). Gradually during the second postnatal week, and sharply throughout the third postnatal week, crypt number increases through crypt fission, with a concomitant expansion of the number of ISCs and Paneth cells.

Hence, for all the above mentioned reasons, abnormal ISC activity should be more noticeable at this period, when major structural and biochemical changes associated with weaning take place (for instance, mice start sampling solid food around P14), helping to explain why the onset of intestinal phenotype in the Mex3a KO mice primarily occurs during the P15-P21 time-window and not in earlier stages.

2. To evaluate the presence of surviving *Lgr5*⁺ ISCs (perhaps due to some mosaicism from rescue from *Mex3a* paralogous members), can the authors use lineage tracing with *Mex3a*KO;*Lgr5*-EGFP-IRES-CreERT2;R26-conditional reporter (e.g. *tdTomato*) to show the turnover and what are the cell types that can be generated from the *Mex3a*KO ISCs?

We agree that this experiment would allow us to follow the fate of the cell types generated from the *Mex3a* KO ISCs and turnover (please see also our response to Reviewer #1, point 10). Unfortunately, we do not have a *Mex3a*;*Lgr5* lineage-tracing ready strain to conduct this experiment. Still, we believe the absence of Paneth and tuft cells, the BrdU pulse-chase experiments and the ultrastructural characterization of enterocytes, all demonstrate that there is a functional effect of *Mex3a* loss in *Lgr5*⁺ ISCs normal renewal.

3. Similar to point #1, with a marked reduction of *Lgr5*⁺ ISCs, how are the organoids sustainable? There is no mention of the longevity of the organoids derived from *Mex3a* knockout crypts in culture - Can they be propagated for as long as the WT crypts? Or can they only be propagated for a limited time? This reflects the functionality of *Lgr5*⁺ ISCs that remain in the organoids.

We described in the Results section that we did not detect differences in the longevity of organoids originating from *Mex3a* KO mice compared with the WT, at least for a period of 3 months of culture. What was strikingly different from the WT organoids was the growth dynamics within each passage. We hypothesize that *Mex3a* KO organoids are sustainable due to the combination of at least two factors: 1) there is essentially a *Lgr5*⁻ cell population capable of initiating organoid growth from the KO crypts that gives rise to *Lgr5*⁺ cells in culture. This type of dynamic cellular transition has been observed in other models, including the *Lgr5*^{DTR/+} mice in which *Lgr5* KO crypts give rise to organoids with *Lgr5*⁺ cells in culture (Tian *et al.* 2011). This is probably linked to the fact that routine organoid culture conditions stimulate conversion of different cells (possibly TA or already committed progenitors) to *Lgr5*⁺ cells due to the presence of Rspodin in the medium; 2) PPAR γ baseline activity is not different between *Mex3a* KO and WT organoids, possibly due to modulation by components of the culture medium. This means that the Wnt pathway counteracting effect observed *in vivo* is possibly lost *ex vivo*. Moreover, culture conditions are enriched in Wnt pathway promoting signals and for that reason we can only see an impact of the PPAR γ pathway when we use agonists. This feature of the PPAR γ signalling pathway might decrease the relevance of *Mex3a* expression in the organoid system. Being an RBP, it actually makes sense because MEX3A is probably involved in rapid switches of target gene expression in response to external signals that are not faithfully recreated in culture (for example, no fluctuations in nutrient availability, no circadian rhythms, among others). These are hypothesis that we will explore in the future, but we believe fall outside the scope of the current work.

4. As the mouse model is a systemic constitutive knockout of *Mex3a*, can the authors perform transplantation of the organoids into a wildtype host to ascertain the ISC-specific phenotype?

We agree this would be an interesting experiment to better clarify the epithelial ISC-specific phenotype. However, to the best of our knowledge transplantation of organoids in a small intestinal context is not well established. Even transplantation into a colonic context is non-trivial and would require optimizing protocols of induced mucosal damage that would be overlong to establish within the timeframe of the revision. Alternatively, we have isolated mesenchymal cells from both WT and KO small intestinal tissue (after villi and crypt removal), established primary cultures, and produced conditioned media. We observed that it was possible to generate and passage both KO and WT organoids in the presence of either KO or WT mesenchymal cell-derived conditioned media. In addition, we also observed no significant differences in the KO mesenchymal tissue concerning the expression level of genes coding for specific growth factors (*Egf*, *Nog*, *Rspo1*, *Wnt2b* and *Wnt3*) suggesting that there is a minimal contribution of the mesenchyme for the observed phenotype.

5. It is difficult to reconcile the contrasting expression patterns of non-CBC-specific stem cell marks *Hopx* and *Lrig1* in *Mex3a* knockout SI (in which they are downregulated) and organoids (in which they are upregulated). What do the authors make of these opposing observations?

The Reviewer raises a relevant point. In fact, by performing qPCR analysis, we could only confirm the decreased expression of *Lrig1*. This suggests that at least this reserve stem cell population is not able to functionally compensate for the loss of *Lgr5*⁺ ISCs *in vivo*. On the other hand, the expression of both markers is clearly detected in WT organoids (in the budding domains) and *Mex3a* KO spheroids (broadly present). In case of the latter, they might to some extent ensure organoid

culture maintenance in early time-points when *Lgr5* expression is decreased. This type of hierarchical plasticity is also observed in WT organoid cultures (Mustata *et al.* 2013; Smith *et al.* 2018).

6. *RNASEQ and qPCR validation performed on only 2 mice per group - I disagree with what the authors said about the "high degree of similarity within genotypes" (line 411). Looking at the heatmap in Fig 6A, the 4 samples are visibly different from one another, and the two most similar samples are a *Mex3a* knockout and WT (two rightmost columns). Between the two *Mex3a* knockouts, the magnitudes of upregulation in the most highly upregulated genes are don't concur. Therefore, the authors should increase the sample size for the transcriptome profiling and qPCR validation as this has ramifications for the later PPAR γ story.*

We agree with the Reviewer (and Reviewer #3, additional comments) and performed RNA-seq analysis in two additional pairs of *Mex3a* KO and WT mice at P16. In order to remove the confounding factor of age, we present the analysis of the three pairs of mice at P16 only (although when we add the P19 pair the results are quite similar). This data is now included in Fig 4. Due to the number of samples and higher degree of similarity, we were able to apply stringent statistical criteria (P value < 0.01 and $-1.5 \geq \text{fold change} \geq 1.5$) having obtained a list of 725 differentially expressed genes. Most of the individual genes, the biological processes involved, and most importantly the signalling pathways altered in the KO mice are the same when compared to our initial dataset, supporting the results obtained, particularly concerning PPAR γ upregulation and significantly reinforcing Wnt signalling pathway downregulation.

7. *The authors need to show positive control PPIB for the ISH experiments to show that there is no global downregulation of all transcripts since *Mex3a* is an RNA binding protein. Relating to this point, the RNA in situ hybridization panels should be supported by qPCR data as well since qPCR allows for quantitative normalization of each tissue to a housekeeping gene and would address whether the lower expression of the stem cell markers are due to a global downregulation of gene expression (e.g. Figs 3, 4, 5F).*

We performed ISH for *Ppib* (positive control) and *DapB* (negative control) in *Mex3a* KO and WT mice intestinal tissues by routine practice to confirm optimal fixation conditions for the RNAscope ISH staining. We did not observe differences between both mice concerning *Ppib* levels (please see Reviewer Figure 2 below). Global downregulation of all transcripts would not be expected as MEX3A is not a "housekeeping" RBP. Additionally, taking into account our previous work and what is described for *C. elegans*, MEX3A might act at the level of translation, thus differences in mRNA expression levels of direct targets might not be detected. We have performed qPCR for the stem cell markers in isolated crypt fractions and the analysis is now included in Fig 2L.

Reviewer Figure 2. ISH for *Ppib* mRNA, a housekeeping gene, in WT and *Mex3a* KO mice.

8. *Can the authors perform the complementary experiment to Fig 7 and culture *Mex3a* knockout organoids in PPAR γ antagonist to see if it can partially rescue the morphological phenotype and increase *Lgr5* expression?*

Treatment of organoids with the specific PPAR γ antagonist SR2595 did not alter organoid growth or self-renewal ability nor *Lgr5* expression, reinforcing that the WT and KO organoids have an equally low level of PPAR γ pathway baseline activity.

Referee #3:

The manuscript by Pereira and colleagues describes how Mex3a controls maintenance of Lgr5-positive intestinal stem cells. The authors show that Mex3a is required for proliferation of ISCs as well as for differentiation of Paneth cells. They conclude that Mex3a negatively regulates PPAR γ protein levels in the crypt cells. They further demonstrate that PPAR γ signaling impairs growth of intestinal organoids. The manuscript is well written and of general interest. There is one point that need to be addressed to strengthen the conclusions the authors draw.

Major point:

Mex3a was ablated in both epithelium and mesenchyme. Therefore, it is possible that the absence of Wnt and other signals from the mesenchymal cells are responsible for the proliferation defects observed in the mutant mice. The expression of Wnts, BMP antagonists and EGF could be evaluated by qPCR analysis of the mesenchymal cells.

We appreciate that the Reviewer considers our manuscript well written and of general interest. We agree with this point. No significant differences were detected in the KO mesenchymal tissue concerning the expression level of genes coding for specific growth factors (*Egf*, *Nog*, *Rspo1*, *Wnt2b* and *Wnt3*), indicating that *Mex3a* ablation does not alter the expression profile of mesenchymal genes relevant for ISCs maintenance.

Minor point:

Lane 466: The authors write: "MEX3A overexpression in Caco-2 cells led to a 70% decrease in PPAR γ protein levels (Figure S6B)." However, based on their quantification, the protein levels are decreased to 70%. These is a 30% decrease only. Here, the authors should formulate the statement clearly.

We respectfully disagree with the Reviewer, as the statement is correct. Quantification of PPAR γ protein levels upon MEX3A overexpression (PPAR γ bar, middle panel) indeed reveals around 70% decrease in protein expression relative to the Caco-2 mock cell line.

Additional comments Referee #3:

I agree that the study has several weak points to fully support the conclusions of the authors. It has also a technical weakness in RNA-seq analysis.

1. All referees requested either cell type specific Mex3a knock-out or evidences that mesenchymal cells express the signals required for ISC maintenance/ Paneth cell differentiation. The authors propose that Mex3a functions in the intestinal epithelium for the maintenance of ISCs. To provide evidences for the epithelium specific functions of Mex3a they used ex vivo organoid assays and measured cell growth and the expression of marker genes. Figures 5A and 5C show that organoids grow slower, and the expression of Axin2, Lgr5, Lyz1 markers is 2 times lower in the KO versus wt organoids at day 2. However, at day 4 both KO and wt organoids express Axin2 (TA cell marker) and Lgr5 (stem cell marker) at the same level (Figure 5E) and organoids are of a similar size as wt (Figure 5B and 5D). Moreover, the expression of TA cell markers Hopx and Lrig1 are strongly induced in KO organoids exposed to the external growth factors. The organoids could be maintained for months in culture. Therefore, proliferation and ISC/ TA marker expression defects in Mex3a KO mice could be rescued by culturing the epithelial cells with appropriate external growth factors. This indicates that either Mex3a is required in the mesenchymal cells of the gut or it is required for differentiation/ maturation of Paneth cells. Based on the data provided in the current manuscript it is not possible to conclude.

Based on the loss of *Lgr5*⁺ ISCs in the *Mex3a* KO and the link with PPAR γ signalling, our hypothesis is that MEX3A is involved in the regulation of ISCs response to external signals. These signals might operate differently in the organoid culture system or might be entirely lost due to the static properties of the culture. Hence, *Mex3a* does not seem required to ensure organoid maintenance (please see our answer to Reviewer #1, point 8). Although we cannot formally exclude that *Mex3a* is required in the mesenchymal compartment due to the constitutive nature of our model, it seems more plausible, also based on the PPAR γ results, that the role of MEX3A is centred on epithelial ISCs (please see our answer to Reviewer #1, point 7). Concerning the second point, the lack of Paneth cells *in vivo* could be due either to an impairment of ISC differentiation into this lineage or to a drastic reduction in the number of ISCs that originate them. We do not see a blockade

in Paneth cell differentiation, as we are still able to detect them *in vivo* although at very low numbers, and in the *Mex3a* KO organoid cultures, although with a delayed appearance. Hence, we favour the second hypothesis, as there is a strong reduction in the number of *Lgr5*⁺ ISCs *in vivo* and *in vitro* (in initial days of culture).

2. Referee 2 (point 8) has proposed a very good functional assay, however, it cannot be performed for the following reason. Ex vivo, neither pparg nor its target genes are significantly upregulated in Mex3a KO compared to wt organoids. PPARγ is not expressed in the crypt compartment in WT conditions. Since KO cells also do not express it in culture this indicates that the external growth factors influence the expression of PPARγ targets in the epithelial cells.

We appreciate the Reviewer detailed analysis and fully agree with it. Still, we performed this experiment and as expected observed no changes in organoid growth or self-renewal ability nor *Lgr5* expression upon treatment with a specific PPAR γ antagonist.

3. Referee 2 (point 6) finds that mex3a KO is closer to wt in one replicate. This is absolutely right. I have not seen whether the authors indicated that RNA-seq was performed using animals of different age. One pair was at P16 and the other at P19. This is the period when the gut is changing so much. It is why, the samples are separated by age and not by genotype. To be precise, the RNA-seq is done on 1 replicate at different time points. They should provide more replicates at least for one stage.

We agree with the Reviewer analysis (please see also our response to Reviewer #2, point 6) and performed RNA-seq analysis in two additional pairs of *Mex3a* KO and WT mice at P16. This data is now included in Fig 4. Most of the individual genes, the biological processes involved, and most importantly the signalling pathways altered in the KO mice are the same when compared to our initial dataset, supporting the results obtained, particularly concerning PPAR γ .

2nd Editorial Decision

13 January 2020

Thank you for the submission of your revised manuscript to our editorial offices. We have now received the reports from the three referees that were asked to re-evaluate your study, you will find below. As you will see, referees #1 and #3 now support the publication of your study in EMBO reports. Referee #2 remains critical, insisting on a conditional KO approach we have agreed as out of scope of the current study. We thus do not require a conditional mouse model. Nevertheless, please address this referees concerns regarding the quality of the panels (the referee indicates are poorly fixed). Moreover, referee #3 has a final point regarding the correlation between BMI1 as a supposedly proximal stem cell marker and region-specific KO phenotypes. Since *Bmi1* has conclusively been shown to be widely expressed throughout all regions of the intestine, this correlation is not valid and needs to be revised accordingly. Please do that. Please also provide a point-by-point-response to these remaining concerns case.

Further, I have these editorial requests:

- Please provide the abstract written throughout in present tense.
- Please move all material and methods information from the Appendix to the main manuscript, and delete these in the Appendix.
- In the author contributions Valdemar Maximo is missing. Please check and provide his contribution.
- Please go again through all the figure legends, INCLUDING those of the Appendix, and make sure that, where applicable, the number "n" for how many independent experiments (biological replicates) were performed is indicated, the bars and error bars (e.g. SEM, SD) are defined, and the test used to calculate p-values is indicated.
- Please go through all the figure legends, INCLUDING those of the Appendix, and assure that all the scale bars are defined in the legend. It seems at least in the Appendix this is not always the case.
- There seems to be no call-out for Figure 4D in the manuscript text. Please check.

- In the diagrams of Figs. 4H and EV4 you show error bars and/or statistical testing (for 4H), although you indicate in the legend that only two replicates are shown. Thus, statistical testing does not make much sense here (with two replicates). Please show these data without statistics and error bars, by showing the two dataset separated (e.g. one bar for each replicate).

- Table 1 is too long to be shown online in the manuscript. This needs to be a dataset (data files that will be linked to the article). Please upload this file as Dataset file, named Dataset EV1. Please add the legends for this datasets as a new TAB to the respective excel file (as first TAB). Please also update the callouts for this file in the manuscript text. Finally, please remove the legend for this table from the manuscript text.

- As it does not contain main data, I suggest to move Table 2 to the Appendix. Please put this table together with its legend into the Appendix file (after the figures), name it Appendix Table S1, include this in the TOC of the Appendix, and update the callouts for this table in the manuscript file. Finally, please remove the legend for this table from the manuscript text.

- Please remove the reviewer access information from the data availability section and make sure that the data gets public upon publication of the study.

- Finally, please find attached a word file of the manuscript text (provided by our publisher) with changes we ask you to include in your final manuscript text, and some queries, we ask you to address. I think you have already addressed these. But please re-check and provide your final manuscript file with track changes, in order that we can see the modifications done.

In addition I would need from you:

- a short, two-sentence summary of the manuscript
- two to three bullet points highlighting the key findings of your study
- a schematic summary figure (in jpeg or tiff format with the exact width of 550 pixels and a height of not more than 400 pixels) that can be used as a visual synopsis on our website.

REFEREE REPORTS

Referee #1 (referee #3 from TEJ submission):

I have read both the revised manuscript and the comments of the authors. I think they have addressed everything satisfactory. I recommend publication of this work.

Referee #2 (referee #1 from TEJ submission):

This paper provides an initial characterization of the Mex3a KO in the gut. The authors are to be commended for their thoughtful responses to the reviews, but the initial critiques by me and others still hold, specifically, that a conditional KO approach is essential in order to make claims about the role of Mex3a in ISCs. Additionally, the quality of the data could be improved, with the tissue being poorly fixed in most of the panels. I do think that the paper would be a contribution to the literature, but it might be more appropriate in a more specialized journal.

Referee #3 (referee #2 from TEJ submission):

Although the lack of a conditional model and/or transplant experiments somewhat dilutes their major conclusions, the authors have done a good job of addressing the major technical concerns - in

particular increasing the biological replicates.

I would hope that such an experienced intestinal biology group would not continue to propagate the myth that *Bmi1* is selectively marking proximal stem cells simply to justify their observed regional differences in the KO phenotype. Multiple studies have conclusively shown that *Bmi1* is broadly expressed in crypts from all regions of the intestine, highlighting the poor fidelity of the reporter models being used by Capecchi's group.

2nd Revision - authors' response

15 January 2020

EMBOR-2019-48938V2

Referee comments:

Referee #1 (referee #3 from TEJ submission)

I have read both the revised manuscript and the comments of the authors. I think they have addressed everything satisfactory. I recommend publication of this work.

We thank the Reviewer for the help improving the manuscript.

Referee #2 (referee #1 from TEJ submission)

*This paper provides an initial characterization of the *Mex3a* KO in the gut. The authors are to be commended for their thoughtful responses to the reviews, but the initial critiques by me and others still hold, specifically, that a conditional KO approach is essential in order to make claims about the role of *Mex3a* in ISCs. Additionally, the quality of the data could be improved, with the tissue being poorly fixed in most of the panels. I do think that the paper would be a contribution to the literature, but it might be more appropriate in a more specialized journal.*

We thank the Reviewer for the comments and the help improving the manuscript. Regarding the comment on the quality of the data, in accordance with the best histopathological practices, whole intestinal tracts were removed immediately after mice euthanasia and placed in 10% neutral buffered formalin fixative for a period of at least 72h to allow optimal tissue penetration. Our processed and stained tissue slides were examined by a pathologist (co-author Leonor David). We do not detect in the reported panels any signs of poor fixation, such as tissue autolysis, heterogeneous staining of cells within the same region or indistinct subcellular structures. On the contrary, and as an example, we can clearly observe nucleoli within nuclei in H&E staining, an indication of good fixation conditions. We believe the Reviewer might be misled by the ISH panels where a green solution was used as counterstain for the red chromogen. Since this is a cytoplasmatic counterstain, the nuclei are not stained, and tissue morphology is not so defined in appearance. On the other hand, this counterstain offers good contrast to the ISH signals, better highlighting them (particularly when they are weaker), which was why we used it.

Referee #3 (referee #2 from TEJ submission)

Although the lack of a conditional model and/or transplant experiments somewhat dilutes their major conclusions, the authors have done a good job of addressing the major technical concerns - in particular increasing the biological replicates.

*I would hope that such an experienced intestinal biology group would not continue to propagate the myth that *Bmi1* is selectively marking proximal stem cells simply to justify their observed regional differences in the KO phenotype. Multiple studies have conclusively shown that *Bmi1* is broadly expressed in crypts from all regions of the intestine, highlighting the poor fidelity of the reporter models being used by Capecchi's group.*

We thank the Reviewer for the comments and the help improving the manuscript. We agree with the Reviewer's view on *Bmi1* expression, even though following the work of Capecchi's group other labs established *Bmi1*-reporter models and still describe a proximal to distal tracing gradient (Tian *et al.* 2011; Yan *et al.* 2012) or only assess proximal small intestine in related experiments (Li *et al.*

2014; Jadhav *et al.* 2017). There might be regional differences in level of expression and/or stem cell activity for this and/or other markers, an issue that is not fully disclosed. Nevertheless, we believe the main reason for the *Mex3a* KO phenotype to be more evident in the distal regions is the difference in turnover rate, which makes these more vulnerable to alterations in ISC numbers.

Accepted

20 January 2020

I am very pleased to accept your manuscript for publication in the next available issue of EMBO reports. Thank you for your contribution to our journal.

YOU MUST COMPLETE ALL CELLS WITH A PINK BACKGROUND ↓
PLEASE NOTE THAT THIS CHECKLIST WILL BE PUBLISHED ALONGSIDE YOUR PAPER

Corresponding Author Name: Bruno Pereira
Journal Submitted to: EMBO Reports
Manuscript Number: EMBOR-2019-48938V2